# Fish-hunting cone snail venoms are a rich source of minimized ligands of the vertebrate insulin receptor

Peter Ahorukomeye[1†], Maria M Disotuar[2†], Joanna Gajewiak[1], Santhosh Karanth[3,4,5], Maren Watkins[1], Samuel D Robinson[1], Paula Flórez Salcedo[2], Nicholas A Smith[6], Brian J Smith[6], Amnon Schlegel[2,3,4,5], Briony E Forbes[7], Baldomero Olivera[1], Danny Hung-Chieh Chou[2], Helena Safavi-Hemami[1,2]*

[1]Department of Biology, University of Utah School of Medicine, Salt Lake City, United States; [2]Department of Biochemistry, University of Utah School of Medicine, Salt Lake City, United States; [3]Molecular Medicine Program, University of Utah, Salt Lake City, United States; [4]Department of Internal Medicine, Division of Endocrinology, Metabolism and Diabetes, University of Utah School of Medicine, Salt Lake City, United States; [5]Department of Nutrition and Integrative Physiology, College of Health, University of Utah, Salt Lake City, United States; [6]La Trobe Institute for Molecular Science, La Trobe University, Melbourne, Australia; [7]Department of Medical Biochemistry, Flinders University, Bedford Park, Australia

**Abstract** The fish-hunting marine cone snail *Conus geographus* uses a specialized venom insulin to induce hypoglycemic shock in its prey. We recently showed that this venom insulin, Con-Ins G1, has unique characteristics relevant to the design of new insulin therapeutics. Here, we show that fish-hunting cone snails provide a rich source of minimized ligands of the vertebrate insulin receptor. Insulins from *C. geographus*, *Conus tulipa* and *Conus kinoshitai* exhibit diverse sequences, yet all bind to and activate the human insulin receptor. Molecular dynamics reveal unique modes of action that are distinct from any other insulins known in nature. When tested in zebrafish and mice, venom insulins significantly lower blood glucose in the streptozotocin-induced model of diabetes. Our findings suggest that cone snails have evolved diverse strategies to activate the vertebrate insulin receptor and provide unique insight into the design of novel drugs for the treatment of diabetes.

*For correspondence:
helena.safavi@utah.edu

†These authors contributed equally to this work

Reviewing editor: Pierre De Meyts,

## Introduction

Insulin is a pancreatic hormone that is critical for glucose homeostasis. Secretion of insulin from pancreatic β-cells triggers the uptake of blood glucose into a variety of tissues, including the liver, skeletal muscle and adipose cells. Impairment of insulin secretion and/or insensitivity to the insulin produced can lead to the disease diabetes mellitus. To date, three major types of diabetes have been recognized: Type one diabetes (T1D) or autoimmune diabetes, Type two diabetes (T2D) and gestational diabetes mellitus (GDM). Daily insulin injections are the only effective treatment for patients with T1D, late-stage T2D and some with GDM (*Mack and Tomich, 2017*). A major limitation of insulin therapy is its delayed action. The native hormone (consisting of an A and B chain connected by disulfide bonds) oligomerizes into a stable hexamer comprising three insulin dimers held together by two central zinc ions (*Adams et al., 1969*). Following subcutaneous injection, the hexamer has to dissociate into the dimer, then monomer, in order to activate the insulin receptor.

**eLife digest** Insulin is a hormone critical for maintaining healthy blood sugar levels in humans. When the insulin system becomes faulty, blood sugar levels become too high, which can lead to diabetes. At the moment, the only effective treatment for one of the major types of diabetes are daily insulin injections. However, designing fast-acting insulin drugs has remained a challenge. Insulin molecules form clusters (so-called hexamers) that first have to dissolve in the body to activate the insulin receptor, which plays a key role in regulating the blood sugar levels throughout the body. This can take time and can therefore delay the blood-sugar control.

In 2015, researchers discovered that the fish-hunting cone snail *Conus geographus* uses a specific type of insulin to capture its prey – fish. The cone snail releases insulin into the surrounding water and then engulfs its victim with its mouth. This induces dangerously low blood sugar levels in the fish and so makes them an easy target. Unlike the human version, the snail insulin does not cluster, and despite structural differences, can bind to the human insulin receptor.

Now, Ahorukomeye, Disotuar et al. – including some of the authors involved in the previous study – wanted to find out whether other fish-hunting cone snails also make insulins and if they differed from the one previously discovered in *C. geographus*. The insulin molecules were extracted and analyzed, and the results showed that the three cone snail species had different versions of insulin – but none of them formed clusters. Ahorukomeye, Disotuar et al. further revealed that the snail insulins could bind to the human insulin receptors and could also reverse high blood sugar levels in fish and mouse models of the disease.

This research may help guide future studies looking into developing fast-acting insulin drugs for diabetic patients. A next step will be to fully understand how snail insulins can be active at the human receptor without forming clusters. Cone snails solved this problem millions of years ago and by understanding how they have done this, researchers are hoping to redesign current diabetic therapeutics. Since the snail insulins do not form clusters and should act faster than currently available insulin drugs, they may lead to better or new diabetes treatments.

Hexamer-to-monomer conversion is a slow process that can lead to a significant delay in glucose control. This limitation has spurred efforts to design insulin analogues with reduced dimerization (and thus oligomerization) rates (*Owens, 2002*). However, despite decades of research the best fast-acting insulin formulations are not truly monomeric and still require 15–90 min to effectively lower blood glucose (*Elleri et al., 2011*); this is because the region involved in dimerization of the insulin molecule, the *C*-terminus of the insulin B chain, is also of critical importance for receptor activation (*Menting et al., 2013*). Until now, removing this region of the B chain in order to generate a fast-acting analog could not be achieved without a near complete loss of biological activity (*Bao et al., 1997*; *De Meyts et al., 1978*). Novel insights for potentially solving this long-standing problem recently came from our discovery of an insulin peptide found in the venom of the fish-hunting cone snail, *Conus geographus* (*Safavi-Hemami et al., 2015*). *C. geographus* belongs to a large genus of predatory marine snails that use their complex venoms for prey capture, defense and competitive interactions (*Olivera, 1997*).

*C. geographus* insulin (Con-Ins G1) was shown to rapidly induce insulin shock (dangerously low blood sugar) in its fish prey (*Safavi-Hemami et al., 2015*; *Robinson and Safavi-Hemami, 2016*). Remarkably, Con-Ins G1 lacks the region of the B chain that is critical for both, dimerization and receptor engagement in human insulin (*Safavi-Hemami et al., 2015*; *Menting et al., 2016*). Despite this, Con-Ins G1 is a potent agonist of the human insulin receptor (*Menting et al., 2016*). Structure-function studies provided a rationale for this conundrum: two residues within the B chain of Con-Ins G1 act as surrogates for the missing *C*-terminus of the B chain of human insulin (*Menting et al., 2016*).

Here, we demonstrate that fish-hunting cone snails have evolved a diverse set of B chain minimized insulins as part of their complex predation strategy. Remarkably, these insulins activate the vertebrate insulin receptor (in fish, mouse and human) but residues previously identified to serve as surrogates for the loss of the B region in Con-Ins G1 are variable. Molecular dynamics simulations reveal the *modus operandi* of these unique molecules. Our findings suggest the evolution of diverse

molecular mechanisms of insulin receptor activation, providing a set of solutions to potentially solve a long-standing problem of designing truly monomeric, fast-acting insulin analogs for the treatment of diabetes.

## Results

### Identification and analysis of venom insulin sequences

We investigated three species of fish-hunting cone snails native to the Central Philippines: *C. geographus* and *C. tulipa* from the Gastridium clade and *C. kinoshitai* from the Afonsoconus clade (*Figure 1A*). Reverse transcription PCR (RT-PCR) combined with whole transcriptome sequencing led to the identification of several distinct insulin sequences in the two cone snail species *C. geographus* and *C. tulipa* (*Safavi-Hemami et al., 2015*). Of these sequences, the venom insulin Con-Ins G1 shared highest similarity with fish insulin and was previously selected for functional and structural characterization (*Safavi-Hemami et al., 2015*; *Menting et al., 2016*). Here, we synthesized and functionally characterized four additional venom insulins from these species with various degrees of sequence divergence: Con-Ins G3 from *C. geographus* and Con-Ins T1A, T1B and T2 from *C. tulipa*. Con-Ins G1 and G3 differ by 32 residues across the entire precursor (22 residues within the A and B chain, *Figure 1—figure supplement 1*) and likely represent paralogs originated by gene duplication (*Safavi-Hemami et al., 2016*). Con-Ins T1A and T1B only differ by three residues across their entire precursor sequence and likely represent allelic variants of the same gene locus. Con-Ins T2 differs at 15 positions across the precursor (13 positions within the A and B chain) and likely originates from a distinct gene locus (*Figure 1—figure supplement 1*). In addition to these sequences, we performed RT-PCR and whole transcriptome sequencing of the venom gland of *C. kinoshitai* leading to the identification of two new insulin sequences, named Con-Ins K1 and Con-Ins K2. The two *C. kinoshitai* insulin sequences differ at 24 positions, 22 of which fall within the A and B chain (*Figure 1—figure supplement 1*). Compared to insulins from *C. geographus* and *C. tulipa*, the C-peptide regions of Con-Ins K1 and Con-Ins K2 are shorter suggesting an insertion/deletion event of 14 amino acids (14 codons, 42 nucleotides) during the evolution of these peptides.

As evident by a highly conserved N-terminal signal sequence, all venom insulins belong to the same gene superfamily of venom gland-specific insulins (*Safavi-Hemami et al., 2016*). Sequence variability is highest within the A and B chain while the signal sequence is conserved (*Figure 1—figure supplement 2*). This juxtaposition between conserved and hypervariable regions is a common characteristic of cone snail venom toxins (*Woodward et al., 1990*).

Cleavage of the A and B chains and post-translational modifications were predicted from precursor sequences based on mass spectrometric sequence information available for venom insulins from *C. geographus* (*Safavi-Hemami et al., 2015*). Unlike the endogenous insulins used by cone snails for insulin signaling (*Safavi-Hemami et al., 2016*), all seven venom insulins exhibit the cysteine framework of vertebrate insulin (with 4 and 2 cysteines in the A and B chain, respectively). Critically, all seven sequences lack the C-terminal region of the B chain involved in dimerization and receptor activation of human insulin (*Figure 1B*). Insulins from *C. geographus* and *C. tulipa* share more sequence similarity to one another than insulins from *C. kinoshitai*, consistent with the close phylogenetic relationship of these two species (*Puillandre et al., 2014*). Con-Ins K1 and K2 have longer A chain C-termini and B chain N-termini and differ in nearly every position from Con-Ins G1 (*Figure 1B*). Notably, venom insulins are distinct from the endogenous signaling insulin expressed in the circumoesophageal nerve ring of cone snails (*Figure 1C*) (*Safavi-Hemami et al., 2016*). This is also evident by the presence of a distinct signal sequence (and distinct 5' and 3' untranslated regions) between the venom insulin gene family and their endogenous homologs (data not shown). All sequences were deposited to GenBank (see Material and methods section for accession numbers).

Overall, venom insulin sequences exhibit pronounced sequence divergence with very few conserved amino acids. The few relatively well conserved amino acids include the first four residues in the A chain, Gly8, Ser9 and Leu18 in the B chain and all six cysteines (see Sequence Logo in *Figure 1*). Strikingly, the two residues (TyrB15 and TyrB20) previously identified to serve as surrogates in Con-Ins G1 for the missing B chain C-terminus of human insulin are only moderately conserved (TyrB15) or hypervariable (TyrB20).

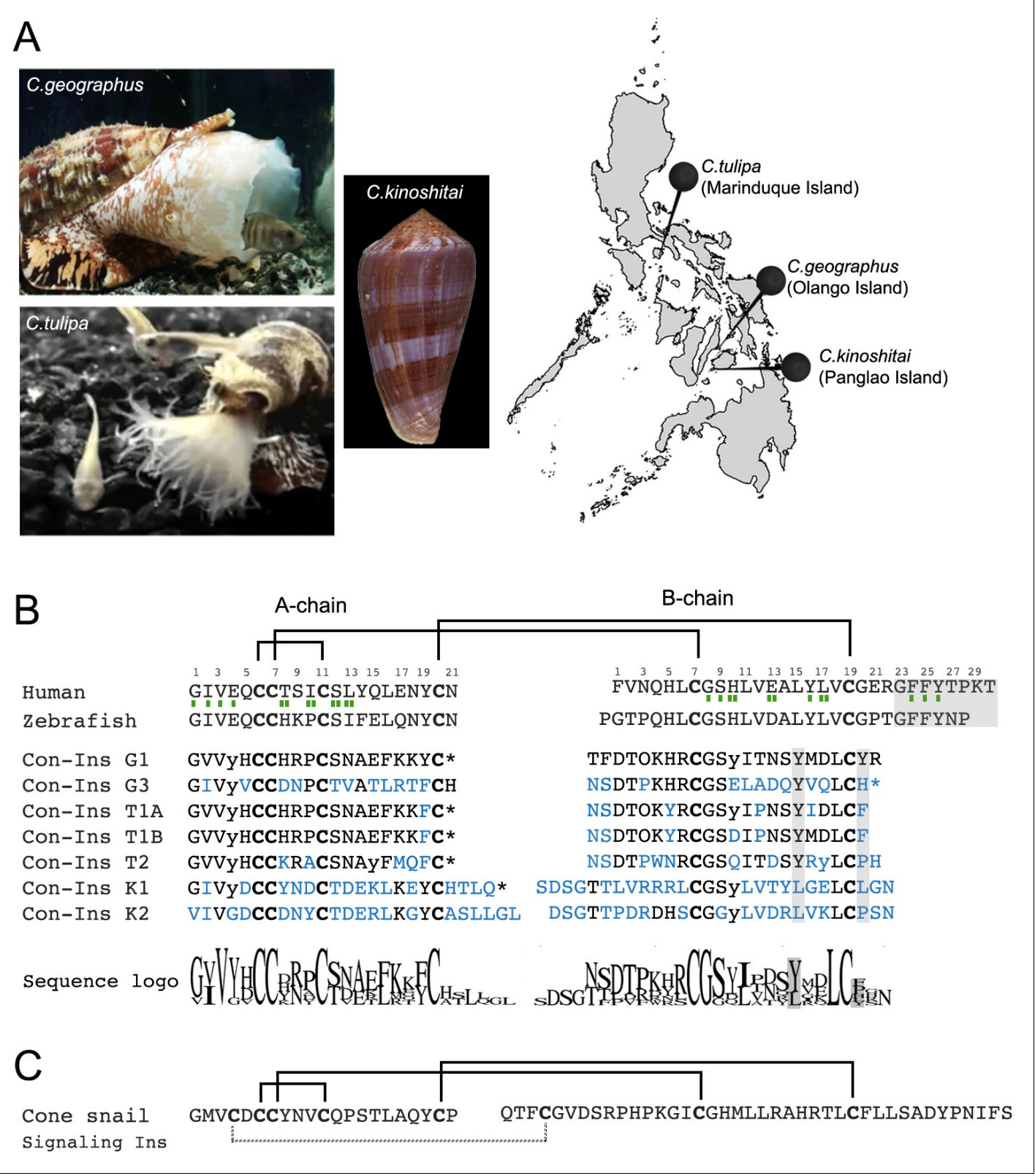

**Figure 1.** Species and insulin sequences analysed in this study. (**A**) Cone snail species analyzed in this study and their collection sites in the Philippines. (**B**) Alignment of the A and B chains of venom insulins with insulin from human and zebrafish. Amino acid numbering for venom insulins was adapted from human insulin (shown on top of sequences). Residues in venom insulins that differ from those in Con-Ins G1 are shown in blue. The sequence logo shows conservation/variability at each position in venom insulins (generated using Geneious vs 11.1.2 software). Cysteines are shown in

*Figure 1 continued on next page*

*Figure 1 continued*

bold and the disulfide connectivity is depicted as connecting lines between cysteines. Post-translational modifications were identified for Con-Ins G1 and Con-Ins G3 (8) and predicted for all other insulins. Modifications are provided in their single letter code. y: y-carboxylated glutamate; O: hydroxyproline, *: C-terminal amidation. Residues of human insulin involved in binding to sites 1 and 2 of the insulin receptor are indicated by a single (site 1) or double (site 2) green rectangle below the hIns sequence (*De Meyts, 2015*). B chain region involved in dimerization and receptor activation in human and fish insulin and positions of residues previously suggested to compensate for the loss of this region in venom insulins are highlighted in gray. (C) The endogenous cone snail signaling insulin was sequenced from the circumoesophageal nerve ring of *C. geographus* (*Safavi-Hemami et al., 2016*). Additional disulfide bond missing in venom insulins is shown as dashed lines between cysteines. Image of *C.tulipa*: courtesy of Jason Biggs.

The online version of this article includes the following figure supplement(s) for figure 1:

**Figure supplement 1.** Full-length precursors sequences of venom insulins characterized in this study.
**Figure supplement 2.** Identity matrix of venom insulins studied here.

## Peptide synthesis

Since its first chemical synthesis in 1963 (*Meienhofer et al., 1963*), insulin has remained challenging to synthesize with the correct intra- and intermolecular disulfide bonds. Here, all novel insulin sequences were successfully synthesized using Fmoc peptide chemistry. Con-Ins G3 and Con-Ins T2 were synthesized using procedures similar to those previously applied to Con-Ins G1 (11). To simplify peptide synthesis for Con-Ins T1A, Con-Ins T1B, Con-Ins K1 and Con-Ins K2, these peptides were synthesized using a selenocysteine replacement strategy in which the intra-molecular disulfide bond in the A-chain is substituted by a diselenide (Sec-Sec) bond (*Safavi-Hemami et al., 2015*). We previously reported that there is no difference in activity between Con-Ins G1 containing a diselenide bond and native Con-Ins G1 (11) and similar observations have been made for other peptides (*Walewska et al., 2009*; *Muttenthaler et al., 2010*). Several different methods were applied for the formation of the first inter-molecular disulfide bridge (*i.e.*, using DMSO, Cu ions and pre-activation of the B-chain with DTNP as recently described by others (*Liu et al., 2014*)). These methods allowed the accumulation of the desired intermediate product containing the first intermolecular disulfide bond in yields of up to 50%. When forming the second intermolecular disulfide bridge between the A and B chain in the presence of iodine we observed the formation of a small amount of an additional disulfide isomer for some venom insulins. This was most pronounced for Con-Ins K1, where two products with the same mass were isolated in almost equimolar ratio. When synthesized with a disulfide instead of a diselenide bond only one final product was observed that was selected for subsequent functional analysis.

## Venom insulins reduce blood glucose in a zebrafish model of diabetes

The presence of insulins with structural similarity to vertebrate insulin in the venoms of these fish-hunting species strongly suggested that these compounds are used to induce hypoglycemic shock in fish prey. Indeed, behavioral observations of *C. geographus* and *C. tulipa* have demonstrated that these species prey on fish and release compounds into the water prior to prey capture (*Olivera et al., 2015*) (*Figure 1A*). While no behavioral data is available for *C. kinoshitai*, phylogenetic analysis places this species within fish-hunters (the majority of cone snail species prey on marine worms and some prey on snails) (*Olivera et al., 2015*). In order to determine whether venom insulins are capable of lowering blood glucose in fish via activation of the insulin receptor, cone snail insulins were tested in the streptozotocin (STZ)-induced model of diabetes in zebrafish (*Safavi-Hemami et al., 2015*). Animals were first rendered hyperglycemic through *i.p.* injection of the β-cell poison STZ (1.5 g/kg) (*Olsen et al., 2010*), and the effects of subsequent injection of each venom insulin were examined. Following STZ treatment, blood glucose levels were significantly elevated from $65.9 \pm 4.8$ mg/dL (n = 11) to $393.3 \pm 10.2$ mg/dL (n = 7). Administration of venom insulins at 65 ng peptide/g body weight significantly lowered blood glucose levels for all venom insulins tested (*Figure 2*) with values ranging from $77.8 \pm 39.8$ mg/dL for Con-Ins T1A ($p<0.0001$, n = 5) to $199.2 \pm 39.8$ mg/dL for Con-Ins T2 ($p<0.0067$, n = 5). For comparison, 65 ng of human insulin (hIns)/g body weight reduces blood glucose to $92.0 \pm 17.4$ mg/dL (*Safavi-Hemami et al., 2015*). This demonstrates that venom insulins are capable of binding to and activating the fish insulin receptor, supporting their biological role in inducing insulin shock in fish prey.

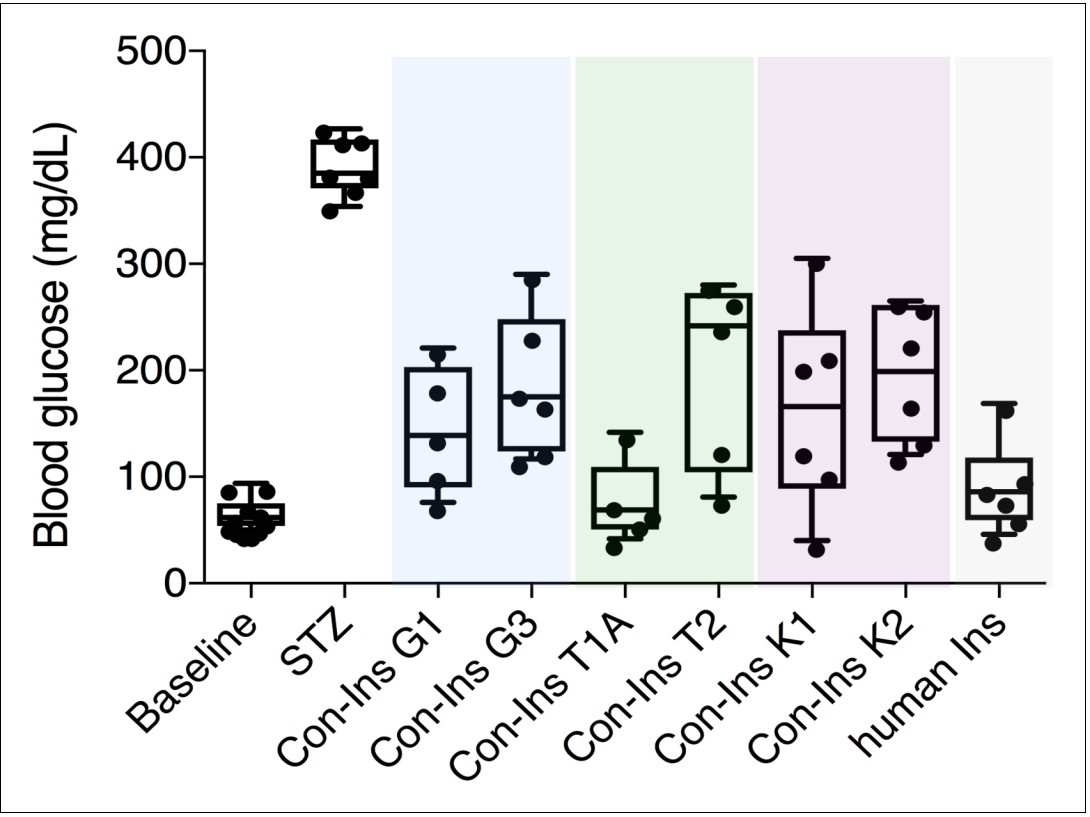

**Figure 2.** Venom insulins significantly lower blood glucose in a zebrafish model of T1D. Insulin activity was determined using the streptozotocin (STZ)-induced model of hyperglycemia. Hyperglycemia was successfully reversed on administration of 65 ng/g of each venom insulin with most significant reduction of blood glucose observed for Con-Ins T1A (p<0.0001, n = 5). Data for human Ins (65 ng/g, n = 6) was replotted from (*Safavi-Hemami et al., 2015*). Data were analyzed in Graphpad Prism software (version 7.0) using unpaired t tests with

*Figure 2 continued on next page*

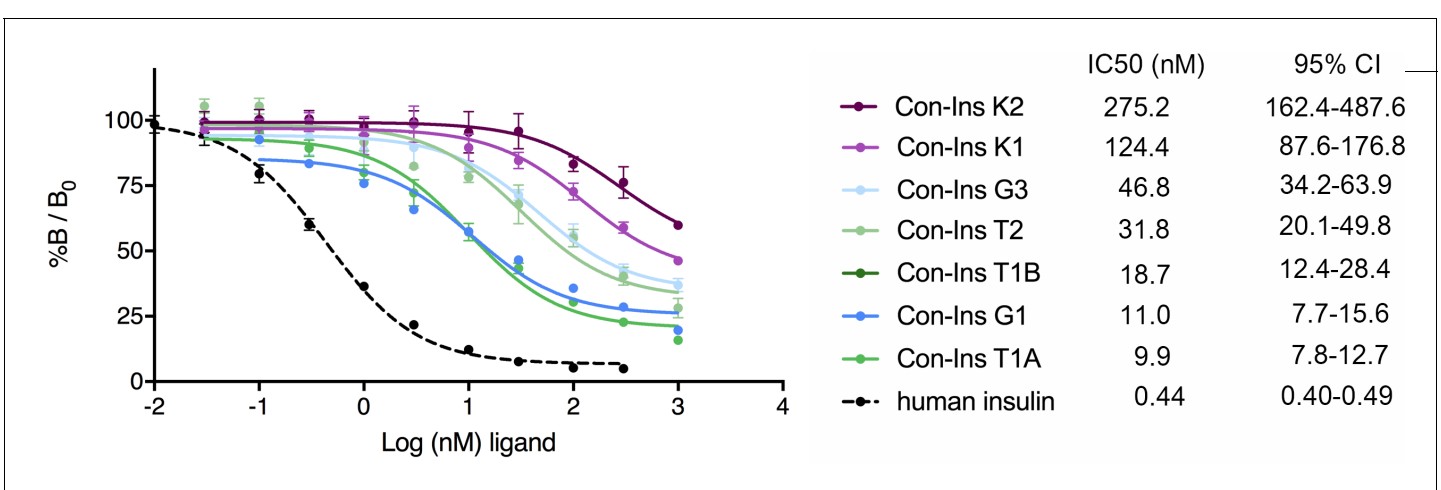

**Figure 3.** Competition binding analysis to hIR-B of venom insulins (n = 2 independent measurements, each comprising three technical replicates). Results are expressed as a percentage of binding in the absence of competing ligand (%$B/B_0$). Plotted values represent means ± s.d. Curve fitting was performed using Prism seven using nonlinear regression (one-site) analysis. IC50 values indicate half-maximal inhibitory concentration and are provided on the right (95% confidence intervals (CI)). Plotted values are provided in Source File 2 (*Figure 3—source data 1*.IR-B.binding).

The online version of this article includes the following source data for figure 3:

**Source data 1.** IR-B.binding: Data plotted in *Figure 3*.

## Venom insulins bind to the human insulin receptor

In order to determine if venom insulins were capable of binding to the human insulin receptor (hIR) the affinity of these compounds for the B isoform of hIR (hIR-B) was determined in competition assays based on the displacement of europium-labeled hIns from solubilized immunocaptured receptors (*Denley et al., 2004*). All seven venom insulins bound to the hIR with affinities ranging from IC50 9.92 nM (7.75–12.68 nM; 95% CI) for the most potent insulin Con-Ins T1A to IC50 275.2 nM (162.2–487.6 nM; 95 % CI) for Con-Ins K2 (n = 2 biological replicates with three technical replicates each; the IC50 indicates half-maximal inhibitory concentration; *Figure 3*). For comparison, the affinity of hIns is ~20 fold higher than the most potent venom insulin Con-Ins T1A [hIns IC50 0.44 nM (0.40–0.49 nM; 95 % CI)].

## Binding of venom insulins to the hIR induces downstream insulin signaling

The ability of venom insulins to activate insulin receptor signaling upon receptor binding was determined using an immunoassay of phosphorylated Akt (pAkt) Ser473 from lysates of mouse NIH 3T3 fibroblast cells overexpressing hIR-B. As previously shown, Con-Ins G1 potently activates the human insulins receptor, albeit at lower potency when compared to human insulin (EC50 of Con-Ins G1 16.28 nM; 95% CI: 7.3–36.4 nM; EC50 of human insulin 1.5 nM; 95% CI: 1.1–2.1 nM, in which the EC50 indicates half-maximal effective concentration) (*Figure 4A and E*). The other insulin identified from *C. geographus*, Con-Ins G3, was significantly less potent at the hIR-B with an EC50 of 242.0 nM, 95% CI: 101.3–578.5 nM). In contrast, all three insulins identified from *C. tulipa* were potently active at the hIR-B with activities comparable to Con-Ins G1 (EC50 of Con-Ins T1A 12.0, 95% CI: 9.7–15.0 nM; EC50 of Con-Ins T1B 12.0 nM, 95% CI: 10.2–13.8 nM; EC50 of Con-Ins T2 15.5 nM, 95% CI: 11.9–20.2 nM) (n = 4 technical replicates; *Figure 4B and E*). *C. kinoshitai* insulins showed variable activity with an EC50 of 30.45 nM (95 % CI 16.9–55.0 nM) for Con-Ins K1 and 373.2 nM (95 % CI 61.6–2262.0 nM) for Con-Ins K2 (*Figure 4C and E*). The ability of Con-Ins T1A and Con-Ins G1 to induce hIR-B phosphorylation (Tyr1150/1151) correlated with downstream receptor activation (Con-Ins T1A and Con-Ins G1 were 10- and 15-times less potent in these assays when compared to hIns; *Figure 4—figure supplement 1*). Overall, the ability of venom insulins to bind to the hIR correlates with downstream signaling activity although some differences can be observed. These include that Con-Ins T2 and Con-Ins K1 have higher and Con-Ins G3 lower activation potency than would be expected from their receptor binding potencies. These observations were not further explored in the current study but could indicate biased signaling of some venom insulins following receptor binding and/or partial receptor antagonism.

As pointed out above, venom insulins lack the region of the B-chain that is known to be important for receptor activation for hIns. The ability of the venom insulins to induce downstream signaling was compared to a B-chain truncated analog of hIns, des-octa peptide insulin (DOI) (*Figure 4D and E*). Of the seven venom insulins tested, five had significantly higher activity at the hIR-B than DOI suggesting the presence of structural motifs that enable receptor activation despite the peptides' lack of the B-chain C-terminal segment.

## The venom insulins Con-Ins G1, Con-Ins T1A and Con-Ins K1 reduce blood glucose in a mouse model of diabetes

Based on their activity against the hIR-B and ability to lower blood glucose in fish, the most active venom insulin from each species was tested in the STZ-induced mouse model of T1D. Animals were rendered hyperglycemic through i.p. injection of STZ. Blood glucose was monitored over the course of 2–3 days prior to administration of human insulin (n = 5 biological replicates) and venom insulins (n = 3 biological replicates for each venom insulin). Given their ~ 10 to 20-fold lower activity at the hIR-B over human insulin, venom insulins were initially injected at 10-times the effective concentration reported for human insulin (1 IU/kg body weight) (*Gupta et al., 2014*). At this concentration Con-Ins G1, Con-Ins T1A and Con-Ins K1 effectively reversed hyperglycemia when measured every 15 min following injection over the course of 125 min (*Figure 5*). Given the severe drop in blood glucose, mice injected with Con-Ins T1A were fed at 90 min post-injection (black arrow). Following this, Con-Ins T1A was also administered at the same concentration of human insulin (equivalent to 1 IU/ kg body weight). At this concentration, Con-Ins T1A effectively lowered blood glucose from

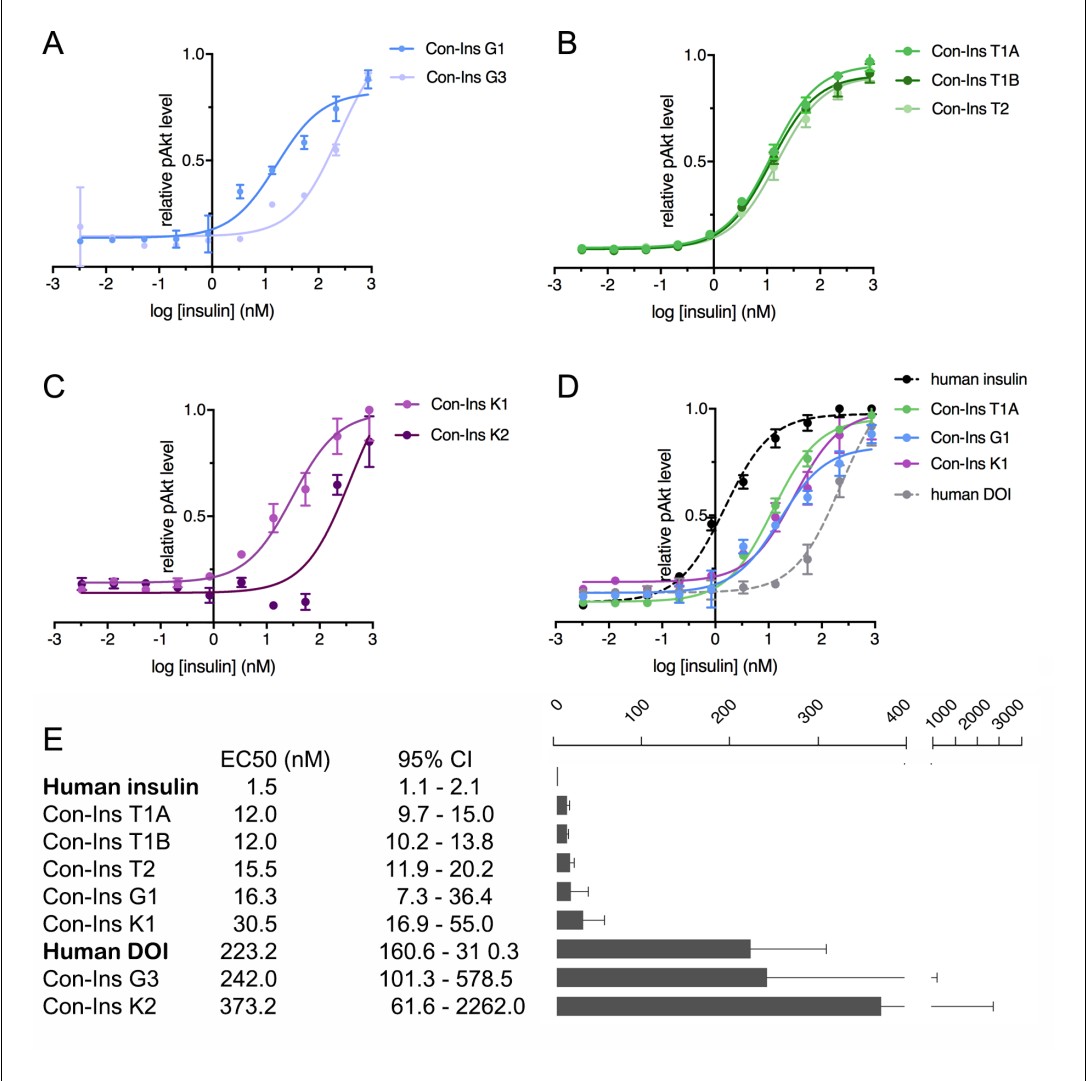

**Figure 4.** Human insulin receptor activation data. Human insulin receptor isoform B (hIR-B) activation (Akt phosphorylation) analysis of venom insulins from (A) *C. geographus*, (B) *C. tulipa* and (C) *C. kinoshitai*. Plotted values represent means ± s.e.m. Curve fitting was performed using Prism seven using nonlinear regression (one-site) analysis. (D) Comparison between the most active venom insulin from each species (Con-Ins T1A, Con-Ins G1 and Con-Ins K1) to native human insulin (dotted black line) and the human des-octa peptide insulin analog (DOI). (E) EC50 (nM) values and 95% confidence intervals (CI) for all insulins tested in this study. Plotted values are provided in Source File 3 (*Figure 4—source data 1*.IR-B.activation).
The online version of this article includes the following source data and figure supplement(s) for figure 4:

**Source data 1.** IR-B.activation: Data plotted in *Figure 4*.
**Figure supplement 1.** Human IR-B autophosphorylation.

541.3 ± 52 mg/dL to 94.3 ± 29 mg/dL over 105 min (*Figure 5C*). Consistent with hIR-B binding and activation data, Con-Ins K1 was less effective at lowering blood glucose at 10-times the concentration of human insulin (*Figure 5D*). When tested at 20-times (equivalent to 20 IU/kg body weight) Con-Ins K1 showed a similar in vivo effect to human insulin.

## Molecular modelling of Con-Ins T1A and Con-Ins K1 identifies residues critical for insulin receptor activation

In order to develop an understanding of the structure and interaction of the most potent venom insulins identified from each species with the insulin receptor, we created a model of Con-Ins T1A and Con-Ins K1 based on the recent crystal structure of Con-Ins G1, in complex with the human insulin micro-receptor (hIR) consisting of the first leucine-rich (L1) (residues 1–154) and the C-terminal

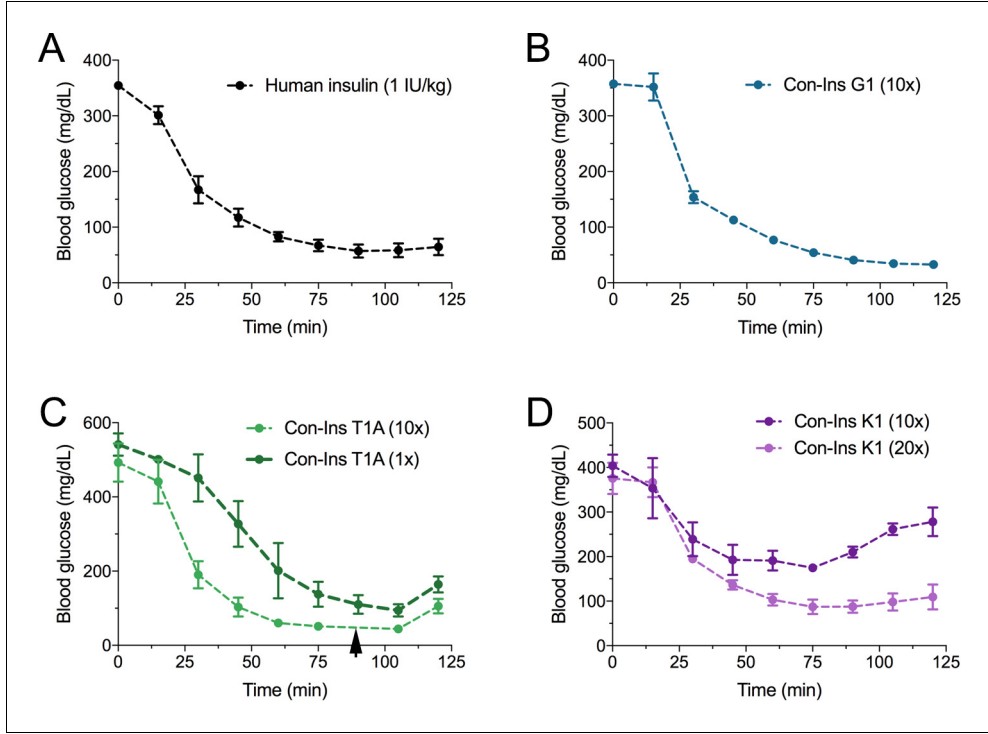

**Figure 5.** The three venom insulins Con-Ins G1, Con-Ins T1A and Con-Ins K1 lower blood glucose in a mouse model of T1D. Insulin activity was determined using the streptozotocin (STZ)-induced model of hyperglycemia. Injection of 0.1 g/kg STZ into adult mice caused hyperglycemia as evident by a high level of blood glucose (~350–580 mg/dL) prior to insulin administration. Hyperglycemia was successfully reversed on administration of 1 unit/kg of human insulin and a 10-times equivalent of Con-Ins G1 and Con-Ins T1A. Given its high potency, mice injected with Con-Ins T1A were fed at 90 min post-injection (black arrow) and Con-Ins T1A was also administered at the same concentration as human insulin. Compared to Con-Ins G1 and Con-Ins T1A, Con-Ins K1 was less potent at 10-times and transiently reduced blood glucose from 404 (±25) mg/dL to 278 (±32) mg/dL. At 20-times Con-Ins K1 showed similar effects to human insulin at 1X. This is consistent with its lower potency at the hIR-B. Plotted data represent mean ±s.e.m (n = 3). Plotted values are provided in Source File 4 (**Figure 5—source data 1**Mouse.STZ. Assay).

The online version of this article includes the following source data for figure 5:

**Source data 1.** Mouse.STZ.Assay: Data plotted in **Figure 5**.

segment (αCT; residues 704–719) of the hIR α-chain (**Menting et al., 2013**; **Menting et al., 2014**) (**Figure 6**, top panel). The model of Con-Ins T1A monomer was stable throughout the 50 ns molecular dynamics (MD) simulation, with notable flexibility of residues TyrB15 and PheB20; the remaining residues exhibited little movement, with the exception of the N-terminal region of the B chain, which exhibited a high degree of motility. The flexibility displayed by the sidechains of TyrB15 and PheB20 correlates with the weak electron density observed for the corresponding residues in the X-ray crystal structure of Con-Ins G1 (11). The sidechains of TyrB15 and PheB20 are locked in position in the model of Con-Ins T1A bound to the hIR over 100 ns (**Figure 6**, middle panel), preferentially adopting a conformation consistent with these residues acting as a surrogate for the receptor-engaging residue PheB24 of the hIns. These positions are identical to positions previously identified in the model of G1 although TyrB20 is replaced by PheB20 in Con-Ins T1A (**Figure 6**, middle panel). Other residues that differ between Con-Ins T1A and G1 differ primarily at the interaction face of L1 and the insulins' corresponding B chains (shown in yellow in **Figure 6**, middle panel). Residues ProB12 and IleB16 of T1A, threonine and methionine in Con-Ins G1, respectively, bind pockets on the surface of L1.

In the model of Con-Ins K1 bound to the hIR, again residues at positions B15 and B20 again appear to play a role in receptor binding. However, aromatic residues present in Con-Ins G1 and T1A at these positions are replaced by leucine in Con-Ins K1. Additionally, and likely compensatorily,

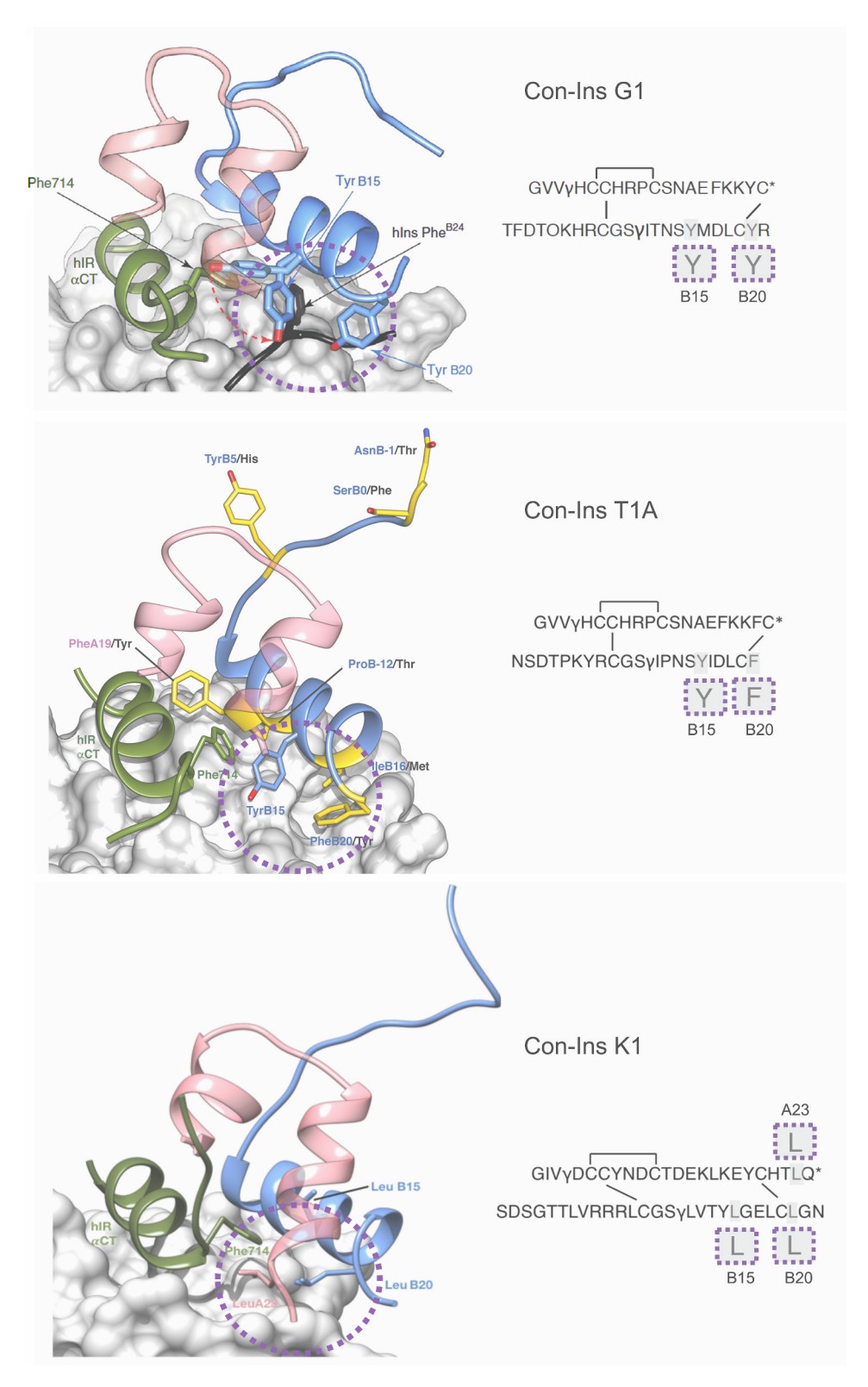

**Figure 6.** Three-dimensional molecular model of Con-Ins G1 (top panel), Con-Ins T1A (middle) and Con-Ins K1 (bottom) in the context of the primary binding site of the human insulin receptor. A and B chains are shown in pink and blue, respectively. hIns residues B22–B27 (in their hIR-bound form) are overlaid in black in the top panel. Sequences of venom insulins are shown below models. Molecular models illustrate how the side chains of B15, B20 and A23 (shown enlarged in sequence representation below models) may act as surrogates for the absence of hIns PheB24 by interacting with αCT

*Figure 6 continued on next page*

*Figure 6 continued*

Phe714. Residues that differ between Con-Ins G1 and Con-Ins T1A are shown in yellow in the model of Con-Ins T1A bound to the receptor (middle panel); labelling of amino acids reflects the residues in T1A (blue) and G1 (black). Modifications are provided in their single letter code. γ: γ-carboxylated glutamate; O: hydroxyproline, *: C-terminal amidation. A chains are transparent for clarity. Top panel was adapted from *Figure 3* in *Menting et al., 2016* (*Menting et al., 2016*).

The online version of this article includes the following source data for figure 6:

**Source data 1.** Three-dimensional molecular model of Con-Ins T1A in the context of the primary binding site of the human insulin receptor.
**Source data 2.** Three-dimensional molecular model of Con-Ins K1 in the context of the primary binding site of the human insulin receptor.
**Source data 3.** .Three-dimensional model of the Con-Ins T1A monomer.

the longer C-terminus of the A chain allows for a continuation of the C-terminal A chain helix orienting LeuA23 towards the hydrophobic core (*Figure 6*, bottom panel). The significant sequence divergence across Con-Ins K1 is compensatory both internally and in maintaining interactions with L1.

Coordinates for MD models are provided as pdb files in the supporting information (*Figure 6*, *Figure 6—source data 1*; *Figure 6—source data 2*; *Figure 6—source data 3*).

## Discussion

Insulin and related peptides (insulin-like peptides, insulin-like growth factors and relaxins) form a large superfamily of peptide hormones (*Shabanpoor et al., 2009*) that is found throughout the animal kingdom (*Piñero-González and González-Pérez, 2011*). In vertebrates, insulin is synthesized in pancreatic β-cells and is the key regulator of carbohydrate and fat metabolism (*Blumenthal, 2010*). Insulin expression has also been detected in the brain where it functions in energy homeostasis and cognition (*Gerozissis and Kyriaki, 2003*). In vertebrates, the primary amino acid sequence, the length of the A and B chains, and the arrangements of cysteines that form disulfide bonds are highly conserved (*Blumenthal, 2010*). In contrast, sequences of invertebrate insulin-like peptides are more variable and are predominantly expressed in neuroendocrine tissues (*Smit et al., 1998*). Most notably, insulins found in mollusks (including cone snails) differ from vertebrate insulins by having one additional disulfide bond between the A and B chain and are larger (*Smit et al., 1998*). Thus, the discovery of an insulin in the cone snail *C. geographus* that unprecedentedly small and shared the disulfide framework with vertebrate insulin was surprising and indicated that this insulin had evolved to rapidly and effectively induce insulin shock in fish prey.

The large majority of the ~800 extant cone snail species prey on worms. Fish- and snail-hunting behaviors are believed to have only evolved in very few clades within this large genus. Within fish-hunters, only a small subset of species expresses insulins that mimic fish insulins (*C. geographus* and *C. tulipa* from the Gastridium clade). These species are known to use other specialized toxins to sedate and disorient their prey prior to capture. Con-Ins G1 was previously suggested to be part of this so-called 'nirvana cabal' that is released into the water to induce hypoglycemic shock and facilitate capture of the incapacitated prey. Here, we show that *C. kinoshitai*, a distantly related fish-hunter, also expresses insulins in its venom. The hunting strategy of this deep-water species has never been documented, but the presence of venom insulins suggests that *C. kinoshitai* may utilize these insulins for prey capture. We show that all seven venom insulins identified from these three fish-hunters are active at the vertebrate receptor, albeit with varying potencies.

Cone snail venom peptides are known for their rapid rate of gene duplications and sequence diversification with nearly no overlap in each toxin repertoire, even between sister species (*Li et al., 2017*). The drastic sequence divergences of venom insulins reported here is consistent with the rapid evolution of cone snail toxins. Notably, while Con-Ins G1 and Con-Ins T1A share sequence similarity with fish insulin, the other five venom insulins exhibit little to no sequence similarity with the fish hormone. Yet, all potently lower blood glucose in a zebrafish model of diabetes suggesting that, in cone snails, diverse strategies have evolved to bind to and activate the fish insulin receptor. *Conus geographus* is one of the largest fish-hunting cone snail species (shell length of ~15 cm) that can produce approximately 50 mg of venom in its long and convoluted venom gland. We have previously determined that venom insulins constitute ~1/25 of the total venom of this species (*Safavi-Hemami et al., 2015*), corresponding to ~2 mg. While it remains to be experimentally determined

how much venom is released into the water during each predation event, if all venom were injected, 2 mg of venom insulin would be sufficient to effectively lower blood glucose in ~85,500 zebrafish at the concentration used in this study (65 ng insulin/g body weight;~23 ng venom insulin per fish).

We further show that three of these insulins (Con-Ins G1, Con-Ins T1A and Con-Ins K1) are capable of lowering blood glucose in a mouse model of T1D demonstrating their in vivo activity in a mammalian model system of the disease.

Notably, each species expresses at least two different insulins with varying sequence divergence. The most pronounced sequence differences within the same species are observed between Con-Ins G1 and Con-Ins G3 from *C. geographus* (11 and 10 residues differ in the A and B chain, respectively). Both insulins lower blood glucose in zebrafish with comparable potencies. However, Con-Ins G3 is more than one order of magnitude less potent against the human IR-B than Con-Ins G1. Similar observations were made for Con-Ins K1 and K2 from *C. kinoshitai* with Con-Ins K2 being ~10 times less active against the human IR-B than Con-Ins K1. These species-specific differences may suggest that cone snails have evolved different insulins to target the insulin receptor in different species of prey. Fish insulin receptors share between ~75–96% identity (for example, zebrafish IR is 96% and 75% identical to the receptor found in the grass carp *Ctenopharyngodon idella* and the green spotted puffer fish *Tetraodon nigroviridis*, respectively). Little is known about the specific diets of fish-hunting cone snails but aquarium observations suggest that they are not restricted to preying on a single fish species (*Cruz and Corpuz, 1978*). Thus, divergent venom insulins may have evolved to allow most efficient capture of different species of prey.

Alternatively, venom insulins may target different isoforms of the insulin receptor within the same prey species. Zebrafish, and other fish species for which genome sequencing data is available, express at least two different isoforms of the insulin receptor (insra and insrb) (*Toyoshima et al., 2008*) that share 75% identity and are differentially expressed in a variety of tissue types (*Tseng et al., 2013*). Humans and other mammals also express two isoforms of the insulin receptor (IR-A and IR-B). However, the mammalian isoforms represent different splice variants of the same gene and their evolutionary origin is distinct from the two isoforms expressed in fish (*Al-Salam and Irwin, 2017*). Testing all venom insulins against different isoforms of the human and fish IR in the future could reveal isoform-specific activity profiles and lead to the generation of isoform-specific insulin receptor probes.

The human and fish insulin receptors are ~65–75% identical and the fish hormone is known to potently activate the mammalian receptor (*Conlon, 2000*). Thus, a venom insulin that evolved to target the insulin receptor in fish has a high chance of also being active at the human receptor. However, activation of the fish and human receptor (and likely all vertebrate insulin receptors) requires a region of the C-terminus of the B chain (*Menting et al., 2014*), particularly the conserved aromatic triplet FFY in position 24–26 (PheB24-PheB25-TyrB26) (*Menting et al., 2014*) that is missing in all venom insulins identified here. Removal of this region from human insulin nearly abolishes its biological activity (*i.e.*, an analog of human insulin missing the eight C-terminal residues of the B chain (DOI) retains less than 0.1% of native insulin bioactivity (*Bao et al., 1997*)). Despite lacking this region, all seven venom insulins bind to and activate the hIR-B and are capable of lowering blood glucose in a zebrafish model of T1D. Besides its role in receptor activation, the C-terminal segment functions in oligomerization of two insulin molecules into a homodimer. Three homodimers form a stable hexamer and insulin is stored and secreted from the β-cells of the pancreas in its hexameric form (*Adams et al., 1969*). Self-association into insulin dimers is stabilized by hydrogen bonds and hydrophobic interactions in a short antiparallel β-sheet of the B-chain C-terminus. Hexamer formation is stabilized by two central zinc ions that form interactions with a histidine at position 10 of the B chains (HisB10) (*Rc et al., 1984*). Removal of the C-terminus of the B chain to prevent dimerization and thus hexamerization have led to fully monomeric insulin analogs (*e.g.*, DOI), however, as mentioned above, these have low receptor binding activities and are poor drug leads.

Of the seven venom insulins tested, five show higher activity at the hIR-B than the B-chain truncated human insulin analog DOI, suggesting the presence of structural motifs that enable receptor activation despite the molecules' lack of the B-chain C-terminus. Identifying these motifs is of significant interest for the design of novel B-chain truncated (thus monomeric) analogs of human insulin that may retain activity at the hIR.

To identify these motifs, we interrogated the mode of binding of Con-Ins T1A and Con-Ins K1 to the human insulin micro-receptor (μIR) (*Menting et al., 2013*). Con-Ins T1A was chosen because of

its superior activity profile over DOI and Con-Ins G1 in all in vitro and in vivo activity assays performed here. Con-Ins K1 was selected based on its higher activity over DOI and its very divergent primary structure compared to other venom insulins and the fish and human hormone. Models of these venom insulins that were stable over 100ns of MD indicated that they contain structural elements that act as surrogates for the B chain aromatic triplet of hIns; inclusive of the key hIR αCT residue Phe714. These residues, Tyr B15 and PheB20 of Con-Ins T1A and, LeuB15, LeuB20 and LeuA23 of Con-Ins K1 occupy space which is otherwise occupied by PheB24 in hIns. Notably, in these models these residues adopt similar rotated conformations as those represented within the model of Con-Ins G1 bound to the same elements of the hIR[3], consistent with the high degree of sequence and modelled structural similarity between Con-Ins T1A and G1, but surprising given the sequence dissimilarity between Con-Ins K1 and G1. However, Con-Ins K1 is significantly less potent than Con-Ins G1 and Con-Ins T1A suggesting that loss of aromatic side chains at positions B15 and B20 results in lower binding affinity and potency at the hIR.

Comparative sequence alignment of a diverse set of venom insulin sequences allows for the interrogation of other residues that may play a role for vertebrate receptor activation but that may not easily identifiable by homology modeling. Amino acids that are conserved in the most potent venom insulins (*i.e.*, Con-Ins T1A, T1B, T2 and G1) and different from the cone snail's own signaling insulin include Glu4 (modified to γ-carboxylated Glu), Lys/Arg9, Ser12 in the A chain and Ser9, Glu/Asp10 (Glu modified to γ-carboxylated Glu), Glu/Asp17 (Glu may be modified to γ-carboxylated Glu) in the B chain. Four of these six residues have previously been shown to play a role for receptor activation by human insulin (GluA4, SerA12, HisB10, LeuB17 in human insulin) (*De Meyts, 2015*). Most notably, mutation of HisB10 to Asp in human insulin leads to a dramatic increase in hIR activation (*Schwartz et al., 1987*). An analog carrying this mutation, insulin X10, was developed as a rapid-acting drug lead for diabetes but was ultimately halted due to its mitogenic properties (*Hansen et al., 2011*). Cone snails appear to have evolved this strategy of introducing a (double)-negatively charged residue at position B10 to enhance vertebrate receptor activation millions of years ago. It seems likely that additional strategies remain to be uncovered. This could include Lys/Arg9 and γ-carboxylated Glu4 in the A chain and Glu/Asp17 in the B chain.

Molecular modeling was performed using the human μIR (*Menting et al., 2013*) that includes structural elements of one of the two known ligand interaction sites on the insulin receptor (site 1 and site 2) (*De Meyts, 2015*; *Whittaker et al., 2008*). Recently published cryo-EM structures of insulin in complex with the human insulin receptor (*Scapin et al., 2018*; *Weis et al., 2018*) may provide opportunities to fully investigate the mode of binding of venom insulins with the hIR and inform on additional structural motifs important for receptor activation. Ultimately, incorporating these structural motifs into DOI or other B chain-truncated analogs of human insulin is likely to lead to the generation of new classes of monomeric insulin analogs for the treatment of diabetes.

Given their streamlined role in prey capture venom insulins may exhibit other advantageous properties that, if uncovered, could inform current drug design efforts. For example, venom insulins may have altered off-rates from the receptor, which would affect the ERK signaling properties and resultant mitogenic activities, or may be more stable in extracellular environments, such as blood. Additionally, it would be interesting to determine if venom insulins lack the negative cooperativity observed for human insulin upon receptor binding (*De Meyts et al., 1978*).

By investigating the venoms of three fish-hunting cone snail species this study characterized seven unique insulin sequences with pronounced sequence divergence and identified key structural elements for hIR activation. Given the large number of species found in the genus *Conus* (~800 species of which ~ 140 species prey on fish) it is likely that additional insulins with unique *modus operandi* at the hIR will be identified from these venoms in the future providing a continuous resource for the design of new insulin analogs inspired by nature.

## Materials and methods

### Specimen collection, RNA preparation and sequencing

All studied specimens were collected in the central Philippines (*Figure 1A*). Specimen identification was initially performed by morphological examination and later verified by sequence analysis of the cytochrome oxidase c subunit 1 (COI) gene as previously described (*Li et al., 2017*). Venom glands

were dissected and stored in RNAlater at −80℃ until further processing. RNA extraction and sequencing of *C. geographus* and *C. tulipa* venom insulins was previously described (*Safavi-Hemami et al., 2015*). Two specimens of *C. kinoshitai* were sequenced in this study. For RT-PCR sequencing, total RNA was extracted from the first specimen using TRIzol Reagent (Life Technologies Corporation) according to the manufacturer's instructions. First-strand cDNA synthesis was performed using AMV reverse transcriptase (Invitrogen) with oligo-dT primer. RT-PCR was performed using the Clontech Advantage 2 PCR Kit. Oligonucleotides were designed based on insulin sequences obtained from the *C. geographus* venom gland transcriptome as previously described (*Safavi-Hemami et al., 2015*) [sense primer: 5′ ACA AGT CAG ATG ACG ACA TC 3′; antisense primer: 5′ ATT CCA T(G,T)C ATG (G,C)GT CAT T 3′]. PCR was carried out for 25 cycles at an annealing temperature of 51℃. To avoid the formation of heteroduplexes, amplicons were diluted 1:5 and subjected to three additional PCR cycles in the presence of fresh buffer, dNTPs, oligonucleotides, and polymerase (*Elleri et al., 2011*). PCR amplicons were gel-purified (Qiagen gel purification kit), cloned into the pGEM-T Easy Vector (Promega), and transformed into *Escherichia coli* (DH10B strain). Plasmids were purified (DNA extraction kit; Viogene-Biotek Corporation) and sequenced at the University of Utah Microarray and Genomic Analysis Core Facility using Sanger DNA sequencing. A total of 10 plasmids was sequenced per species. Sequences represented by at least two clones were considered for subsequent studies.

For whole transcriptome sequencing, total RNA extraction was performed using the Direct-zol RNA extraction kit (Zymo Research, Irvine, CA, USA), with on-column DNase treatment, according to the manufacturer's instructions. cDNA library preparation and sequencing was performed by the University of Utah High Throughput Genomics Core Facility. Briefly, total RNA quality and quantity were first determined on an Agilent 2200 TapeStation (Agilent Technologies). A dual-indexed library was constructed with the Illumina TruSeq Stranded mRNA Sample Prep Kit with oligo (dT) selection and an average insert size of approximately 150 bp. The library was validated on an Agilent 2200 TapeStation and using a qPCR assay (Kapa Biosystems Library Quantification Kit for Illumina), and was multiplexed in a batch of 6 samples. 125 cycle paired-end sequencing was performed on an Illumina HiSeq2000 instrument (San Diego, CA, USA) at an 80% standard cluster density. Adapter trimming of de-multiplexed raw reads was performed using fqtrim (v0.9.4 Release, available online: http://doi.org/10.5281/zenodo.20552), followed by quality trimming and filtering using prinseq-lite (*Schmieder and Edwards, 2011*). Error correction was performed using the BBnorm ecc tool, part of the BBtools package (open source software, Joint Genome Institute). Trimmed and error-corrected reads were assembled using Trinity (version 2.2.1) (*Haas et al., 2013*) with a k-mer length of 31 and a minimum k-mer coverage of 10. Assembled transcripts were annotated using a blastx search (*Altschul et al., 1990*) (E-value setting of 1e-3) against a combined database derived from UniProt, Conoserver (*Kaas et al., 2012*), and an in-house cone snail venom transcript library. An in-house script was used to extract putative toxin transcripts (including the venom insulin gene family), trim to open-reading frame, and discard redundant and partial sequences. Following assembly, venom insulin transcripts were manually examined using the Map-to-Reference tool of Geneious, version 8.1.7 (46).

All seqeucences characterized here have been deposited into the GenBank Nucleotide Database (Accession Numbers: Con-Ins G1: AJD85832; Con-Ins G3: AJD85820; Con-Ins T1A: KP268600; Con-Ins T1B: KP268611; Con-Ins T2: MH879035; Con-Ins K1: MH879033; Con-Ins K2: MH87903).

## Peptide synthesis

Venom insulins were synthesized using solid phase peptide synthesis followed by reversed-phase chromatography and mass spectrometry to verify the identity of all synthetic peptides. Two approaches were used for peptide synthesis: Con-Ins G1, Con-Ins G3 and Con-Ins T2 were synthesized following procedures similar to the method previously described for Con-Ins G1 (*Menting et al., 2016*). Con-Ins T1A, Con-Ins T1B, Con-Ins K1 and Con-Ins K2 were synthesized using a selenocysteine replacement strategy similar to what was previously described for sCon-Ins G1 (*Safavi-Hemami et al., 2016*8). A detailed description of all methods used for peptide synthesis, purification, verification and quantification is provided in Appendix.

### Insulin receptor activation assays

To determine the extent of insulin signaling induced by the different venom insulins, pAkt Ser473 levels were measured in a mouse fibroblast cell line, NIH 3T3, overexpressing human IR-B (a gift from A. Morrione, Thomas Jefferson University). Cells were authenticated by western blotting to assess their level of IR expression compared with that of parent 3T3 cells: the NIH 3T3 cells showed an approximately ten-fold-higher level of expression than that of the parent. Cell lines are tested for mycoplasma contaminations every six months. NIH 3T3 cells were cultured in DMEM (Thermo Fisher Scientific) with 10% FBS, 100 U/mL penicillin-streptomycin (Thermo Fisher Scientific) and 2 µg/mL puromycin (Thermo Fisher Scientific). For each assay, 40,000 cells per well were plated in 96-well plates with culture medium containing 1% FBS. 24 hr later, 50 µL of insulin solution in no FBS media ranging 0.86 µM - 0.82 pM was pipetted into each well following removal of the original medium. After a 30 min treatment, the insulin solution was removed, and the level of intracellular pAkt Ser473 was measured using the HTRF pAkt Ser473 kit (Cisbio) according to the manufacturer's instructions. Briefly, the cells were first treated with cell lysis buffer (50 µL per well) for 1 hr under mild shaking. 16 µL of cell lysate was then added to 4 µL of detecting reagent in a white 384-well plate. After a 4 hr incubation, the plate was read in a Synergy Neo plate reader (BioTek), and data were processed according to the manufacturer's protocol. Mean EC50 values and their 95% confidence intervals were calculated (using GraphPad Prism, version 7) after curve fitting with a nonlinear regression (one-site) analysis.

### Insulin-receptor binding

Competition binding assays were performed with solubilized immunocaptured hIR (isoform B) with europium-labeled human insulin and increasing concentrations of hIns or venom insulin peptides, as previously described (Denley et al., 2004). Time-resolved fluorescence was measured with 340 nm excitation and 612 nm emission filters with a Polarstar Fluorimeter (BMG Labtech). Mean IC50 values and their 95% confidence intervals were calculated with the statistical software package in GraphPad Prism (version 7) after curve fitting with nonlinear regression (one-site) analysis.

### Insulin receptor activation assays

IR-B phosphorylation (Tyr1150/1151) following insulin treatment was measured in a mouse fibroblast cell line, NIH 3T3, overexpressing human IR-B (a gift from A. Morrione, Thomas Jefferson University). NIH 3T3 cells were cultured in DMEM (Thermo Fisher Scientific) with 10% FBS, 100 U/mL penicillin-streptomycin (Thermo Fisher Scientific) and 2 µg/mL puromycin (Thermo Fisher Scientific). For each assay, 45,000 cells per well were plated in 96-well plates with culture medium containing 1% FBS. 24 hr later, 50 µL of insulin solution in no FBS media ranging 1 µM - 0.82 pM was pipetted into each well following removal of the original medium. After a 15 min treatment, the insulin solution was removed, and the level of intracellular Tyr1150/1151 was measured using the HTRF Phospho-IR beta (Tyr1150/1151) kit (Cisbio) according to the manufacturer's instructions. Briefly, the cells were first treated with cell lysis buffer (50 µL per well) for 1 hr under mild shaking. 16 µL of cell lysate was then added to 4 µL of detecting reagent in a white 384-well plate. After a 4 hr incubation, the plate was read in a Synergy Neo plate reader (BioTek), and data were processed according to the manufacturer's protocol. Mean EC50 values and their 95% confidence intervals were calculated (using Graph-Pad Prism, version 7) after curve fitting with a nonlinear regression (one-site) analysis.

### Reversal of STZ-induced hyperglycemia in adult zebrafish

STZ assays were performed on adult zebrafish (strain AB). Studies were approved by the University of Utah Institutional Animal Care and Use Committee. Adult fish of 10–12 months in age and an average weight of 360 ± 80 mg were injected i.p. with 1.5 g/kg STZ (Sigma Aldrich) to cause hyperglycemia. Following STZ injection, animals were fasted for 40 hr and then injected with venoms insulins at 65 ng/g. This concentration was previously shown to be effective when using human insulin and the venom insulin Con-Ins G1 (8). Blood glucose in mg/dL was measured 110 min later with a Bayer Contour meter. Data were analyzed in GraphPad Prism (version 7) using unpaired t tests with Welch's correction.

## Reversal of STZ-induced hyperglycemia in mice

STZ assays on mice were approved by the University of Utah Institutional Animal Care and Use Committee. Adult male mice (CBA/CaJ and C57BL/6J strain) between 12–21 weeks of age with an average weight of 23–37.1 g were injected i.p. with either one injection of 0.15 g/kg STZ (Sigma Aldrich) or one injection of 0.1 g/kg STZ followed by a second dose of 0.05 g/kg STZ after 3 days. Following STZ injections, blood glucose was monitored for 2–3 days until animals became hyperglycemic (blood glucose 350–580 mg/dL). Animals were fasted for 4–6 hr and then injected with human insulin (1 IU/kg, 27.5 IU/mg, Sigma Aldrich) (Gupta et al., 2014) and venom insulins at one-time (1X), ten-times (10X) or twenty-times (20X) the dose of human insulin. Blood glucose was measured (in mg/dL) every 15 min for 125 min following insulin injections using a Bayer Contour meter. Data were analyzed in Prism GraphPad software (version 7.0).

## Homology modeling

Models of Con-Ins T1A and Con-Ins K1 in complex with the IR L1 module (residues His1 to Glu154) and the IR αCT segment (residues Thr704 to Ser719 of the IR-A isoform) were created with MODELLER (v9.16) (Webb and Sali, 2016), with templates of the crystal structure of Con-Ins G1 (PDB 5JYQ (Menting et al., 2016)) and the crystal structure of the IR site one components in complex with hIns (PDB 4OGA(Menting et al., 2014)). Due to the sequence dissimilarity of the templates used, helical and distance restraints were applied to the C-terminus of the A-chain and to LeuB20, respectively, to orient these residues towards the hydrophobic interior. All models included the post-translational modifications of Con-Ins T1A and Con-Ins K1 and a single N-linked N-acetyl-d-glucosamine residue at each of the IR residues Asn16, Asn25, Asn111, Asn215 and Asn255 (Sparrow et al., 2008).

## Molecular dynamics

Molecular dynamics (MD) simulations were conducted as previously described. Briefly, simulations used GROMACS (v5.1.2) (Abraham et al., 2015) with the CHARMM36 force field (Best et al., 2012; Guvench et al., 2011), and were initiated with the models of the Con-Ins T1A-IR and Con-Ins K1-IR complex that had the lowest MODELLER objective function. Ionizable residues were assumed to be in their charged state. Each system was solvated using the TIP3P water model in a cubic box extending 10 Å beyond all atoms. Sodium and chloride ions were added to neutralize the system and provide an ionic strength of 0.1 M. The protein and solvent (including ions) were coupled separately with velocity rescaling to a thermal bath at 300 K applied with a coupling time of 0.1 ps. All simulations were performed with a single nonbonded cutoff of 12 Å, the Verlet neighbor searching cut-off scheme was applied with a neighbor-list update frequency of 25 steps (50 fs); the time step used in all the simulations was two fs. Periodic boundary conditions were used with the particle-mesh Ewald method to account for long-range electrostatics. All bond lengths were constrained with the P-LINCS algorithm (Hess, 2008). Simulations consisted of an initial minimization followed by 50 ps of MD with all protein atoms restrained. After positionally restrained MD, the simulations were continued without restraints for a further 100 ns.

## Acknowledgements

The authors thank Jean E Rivier, Judit Erchegyi and Charleen Miller (Sentia Medical Sciences) for peptide synthesis, Noel Saguil and Jose Arbasto for assistance with specimen collection, Carlie ADelaine and Govinda Poudel for assistance with receptor binding assays and the High Throughput Genomics Core Facility, the Mass Spectrometry and Proteomics Core Facility and the DNA Sequencing Core Facility at the University of Utah for DNA and peptide sequencing. This work was supported in part by a Margolis Foundation grant (to HS-H), a JDRF Innovative Research grant (to HS-H), a National Institutes of Health grant 1RO1GM122869 (to HS-H), an Australian Government Research Training Program (RTP) scholarship (to NAS) and a Utah Science and Technology Initiative (USTAR, to DH-CC).

## Additional information

### Competing interests

Joanna Gajewiak, Santhosh Karanth, Samuel D Robinson, Brian J Smith, Briony E Forbes, Danny Hung-Chieh Chou: author of the following patent application: WO2016172269A3 entitled "Insulin analogs having shortened B chain peptides and associated methods". The other authors declare that no competing interests exist.

### Funding

| Funder | Grant reference number | Author |
| --- | --- | --- |
| National Institute of General Medical Sciences | 1RO1GM122869 | Helena Safavi-Hemami |
| Juvenile Diabetes Research Foundation United States of America | 1-INO-2017-441-A-N | Helena Safavi-Hemami |
| Margolis Foundation | | Helena Safavi-Hemami |
| Australian Government Research Training Program | | Nicholas A Smith |
| Utah Science Technology and Research | | Danny Hung-Chieh Chou |

The funders had no role in study design, data collection and interpretation, or the decision to submit the work for publication.

### Author contributions

Peter Ahorukomeye, Methodology, Writing—original draft, Writing—review and editing; Maria M Disotuar, Joanna Gajewiak, Methodology, Writing—review and editing; Santhosh Karanth, Formal analysis, Methodology, Writing—review and editing; Maren Watkins, Samuel D Robinson, Investigation, Methodology, Writing—review and editing; Paula Flórez Salcedo, Visualization, Methodology, Writing—original draft, Writing—review and editing; Nicholas A Smith, Resources, Data curation, Visualization, Methodology, Writing—original draft, Writing—review and editing; Brian J Smith, Resources, Investigation, Methodology, Writing—review and editing; Amnon Schlegel, Resources, Supervision, Investigation, Methodology, Writing—review and editing; Briony E Forbes, Conceptualization, Resources, Writing—review and editing; Baldomero Olivera, Conceptualization, Resources, Supervision, Methodology, Writing—review and editing; Danny Hung-Chieh Chou, Conceptualization, Resources, Formal analysis, Supervision, Funding acquisition, Investigation, Visualization, Methodology, Writing—original draft, Writing—review and editing; Helena Safavi-Hemami, Methodology

### Author ORCIDs

Amnon Schlegel (iD) http://orcid.org/0000-0003-4060-2252
Helena Safavi-Hemami (iD) https://orcid.org/0000-0002-1984-2721

### Ethics

Animal experimentation: This study was performed in strict accordance with the recommendations in the Guide for the Care and Use of Laboratory Animals of the National Institutes of Health. All of the animals were handled according to approved institutional animal care and use committee (IACUC) protocols of the University of Utah. The protocol was approved on July 28, 2017 (Permit Number: 17-08002).

### Decision letter and Author response

Decision letter https://doi.org/10.7554/eLife.41574.sa1
Author response https://doi.org/10.7554/eLife.41574.sa2

# Additional files

## Supplementary files
• Transparent reporting form

## Data availability

All data generated or analysed during this study are included in the manuscript and supporting files. Source data files have been provided for Figures 1, 2, 3, 4 and 5. The sequences described in this publication have been deposited with GenBank.

The following datasets were generated:

| Author(s) | Year | Dataset title | Dataset URL | Database and Identifier |
|---|---|---|---|---|
| Ahorukomeye P, Disotuar MM, Gajewiak J | 2019 | Fish-hunting cone snail venoms are a rich source of minimized ligands of the vertebrate insulin receptor | https://www.ncbi.nlm.nih.gov/nuccore/MH879033 | GenBank, MH879033 |
| Ahorukomeye P, Disotuar MM, Gajewiak J, Karanth S, Watkins M, Robinson SD, Flórez Salcedo P, Smith NA., Smith BJ, Schlegel A., Forbes BE, Olivera BM, Hung- Chieh Chou D, Safavi-Hemami H | 2019 | Fish-hunting cone snail venoms are a rich source of minimized ligands of the vertebrate insulin receptor | https://www.ncbi.nlm.nih.gov/nuccore/MH879034 | GenBank, MH8790 34 |
| Ahorukomeye P, Disotuar MM, Gajewiak J, Karanth S, Watkins M, Robinson SD, Flórez Salcedo P, Smith NA., Smith BJ | 2019 | Fish-hunting cone snail venoms are a rich source of minimized ligands of the vertebrate insulin receptor | https://www.ncbi.nlm.nih.gov/nuccore/MH879035 | GenBank, MH879035 |

The following previously published datasets were used:

| Author(s) | Year | Dataset title | Dataset URL | Database and Identifier |
|---|---|---|---|---|
| Safavi-Hemami H, Gajewiak J, Karanth S, Robinson SD, Ueberheide B, Douglass AD, Schlegel A, Imperial JS, Watkins M, Bandyopadhyay PK, Yandell M, Li Q | 2015 | Specialized insulin is used for chemical warfare by fish-hunting cone snails | https://www.ncbi.nlm.nih.gov/protein/AJD85832 | GenBank, AJD85832 |
| Safavi-Hemami H, Gajewiak J, Karanth S, Robinson SD, Ueberheide B, Douglass AD, Schlegel A, Imperial JS, Watkins M, Bandyopadhyay PK, Yandell M, Li Q, Purcell AW, Norton RS, Ellgaard L, Olivera BM | 2015 | Specialized insulin is used for chemical warfare by fish-hunting cone snails | https://www.ncbi.nlm.nih.gov/protein/AJD85820 | GenBank, AJD85820 |
| Safavi-Hemami H, | 2015 | Specialized insulin is used for | https://www.ncbi.nlm. | GenBank, |

| Gajewiak J, Karanth S, Robinson SD, Ueberheide B, Douglass AD, Schlegel A, Imperial JS, Watkins M, Bandyopadhyay PK, Yandell M, Li Q, Purcell AW, Norton RS, Ellgaard L, Olivera BM | | chemical warfare by fish-hunting cone snails | nih.gov/nuccore/KP268600 | KP268600 |
|---|---|---|---|---|
| Safavi-Hemami H, Gajewiak J, Karanth S, Robinson SD, Ueberheide B, Douglass AD, Schlegel A, Imperial JS, Watkins M, Bandyopadhyay PK, Yandell M, Li Q, Purcell AW, Norton RS, Ellgaard L, Olivera BM | 2015 | Specialized insulin is used for chemical warfare by fish-hunting cone snails | https://www.ncbi.nlm.nih.gov/nuccore/KP268611 | GenBank, KP268611 |

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

## Appendix 1

### Peptide synthesis

General approach to cleavage and purification of A-chain of Con-Ins T1A and T1B, Con-Ins K1 and K2

A-chains of the insulins were synthesized on 0.05 mmol scale using an Apex 396 automated peptide synthesizer (AAPPTec; Louisville, KY), applying standard solid-phase Fmoc (nine fluorenylmethyloxy-carbonyl) protocols using Fmoc-Cys(Trt) Rink Amide MBHA (0.31 mmol/g load; Peptides International, Louisville, KY) for [$^{A6,11}$Sec, $^{A7}$C(Acm)]Con-Ins T1, Fmoc-Gln(Trt) Rink Amide MBHA (0.25 mmol/g load; Peptides International, Louisville, KY) for [$^{A6,11}$Sec, $^{A7}$C(Acm)]Con-Ins K1and Fmoc-Leu-Wang resin (0.33 mmol/g load; Peptides International, Louisville, KY) for [$^{A6,11}$Sec, $^{A7}$C(Acm)] Con-Ins K2.

A-chain was cleaved from about 100 mg resin by 1.5 hr treatment with 1 mL of enriched Reagent K: 2 mL TFA (Fisher Scientific, Fair Lawn, NJ), 56 µL H2O, 12 mg 2,2-dithiobis(5-nitropyridine) (DTNP; Aldrich; Saint Louis, MO), and 168.5 mg phenol(Acros Organics/Fisher Scientific, Fair Lawn, NJ), followed by addition of 25 µL thioanisole (Aldrich; Saint Louis, MO) and mixing of the reaction for another 1.5 hr. The cleavage mixture was filtered and precipitated with 10 mL of cold methyl-*tert*-butyl ether (MTBE; Fisher Scientific, Fair Lawn, NJ). The crude peptide was precipitated by centrifugation at 7,000 × g for 6 min and washed once with 10 mL cold MTBE. To induce the intramolecular diselenide bond formation (Sec$^{A6}$ to Sec$^{A11}$), the washed peptide pellet was dissolved in 50% acetonitrile (ACN, Fisher Scientific; Fair Lawn, NJ) (vol/vol) in water and 2 mL of 100 mM dithiotreitol (DTT, EMD Chemicals, Gibbstown, NJ) in 1 mL 0.2 M Tris·HCl (Sigma, St Louis, MO) containing 2 mM EDTA (Mallinckrodt, St. Louis, MO), pH 7.5, 1 mL of water was added and vortexed gently, and the reaction was allowed to proceed for 2 hr. It was then quenched with 8% formic acid (vol/vol) (Fisher Scientific, Fair Lawn, NJ), diluted with 0.1% TFA (vol/vol) in water, and purified by reversed-phase (RP) HPLC using a semi-preparative C18 Vydac column (218TP510, 250 × 10 mm, 5 µm particle size; Grace, Columbia, MD) eluted with a linear gradient ranging from 10% to 40% solvent B in 30 min at a flow rate 4 mL/min. The HPLC solvents were 0.1% (vol/vol) TFA in water (solvent A) and 0.1% TFA (vol/vol) in 90% aqueous ACN (vol/vol) (solvent B). The eluent was monitored by measuring absorbance at 220 and 280 nm. Purity of the peptide was assessed by analytical C18 Vydac RP-HPLC (218TP54, 250 × 4.6 mm, 5 µm particle size, Grace, Columbia, MD) using a linear gradient ranging from 15% to 45% of solvent B in 30 min with a flow rate 1 mL/min. [$^{A6,11}$Sec]-Con-Ins T1was quantified by HPLC using A-chain of Con-Ins G1 as a reference peptide. 0.86 mg of [$^{A6,11}$Sec]-Con-Ins T1 was prepared out of 109 mg of the resin, 0.63 mg [$^{A6,11}$Sec]Con-Ins K1 out of 112 mg of the resin and 0.32 mg [$^{A6,11}$Sec]Con-Ins K2 from 91 mg of the resin was prepared. The mass of the peptides was confirmed by electrospray ionization (ESI)-MS using either Waters Micromass Q-ToF-2 or Bruker Maxis II ETD at the Mass Spectrometry and Proteomics Core Facility at the University of Utah. Theoretical mass calculations were made using ChemDoodle by iChemLabs, LLC and http://www.chemcalc.org/

| Peptide | Purity | Yield | Calculated Monoisotopic [M] | Determined Monoisotopic [M] |
|---|---|---|---|---|
| [$^{A6,11}$Sec, $^{A7}$C(Acm)]Con-Ins T1(A and B) | 99% | 2.6% | 2500.92 | 2500.91 |
| [$^{A6,11}$Sec, $^{A7}$C(Acm)]Con-Ins K1 | 80% | 2.8% | 3029.09 | 3029.08 |
| [$^{A6,11}$Sec, $^{A7}$C(Acm)]Con-Ins K2 | 88% | 1.1% | 2987.09 | 2987.18 |

### General approach to cleavage and purification of B-chains of Con-Ins T1A and T1B, Con-Ins K1 and K2

B-chains of the insulins were also synthesized on 0.05 mmol scale using an Apex 396 automated peptide synthesizer applying standard solid-phase Fmoc protocols, using Fmoc-Phe-Wang resin (0.61 mmol/g load; Peptides International, Louisville, KY) for [$^{B9}$C(Acm); $^{B21}$C(Npys)]Con-Ins T1A and T1B and Fmoc-Asn(Trt) Wang resin (0.51 mmol/g load; Peptides International, Louisville, KY) for [$^{B13}$C(Acm)]Con-Ins K1 and [$^{B12}$C(Acm);$^{B24}$C(Npys)]Con-Ins K2.

B-chains (except for [B13C(Acm)]Con-Ins K1, for which standard protocol described in *Proceedings of the National Academy of Sciences of the United States of America* **112**, 1743–1748 (2015) was used] were cleaved from ~100 mg resin by 3 hr treatment with 1 mL of DTNP containing cleavage cocktail: 1 mL TFA, 30 mg DTNP, 25 uL TIS and 25 uL of $H_2O$. Peptides were subsequently filtered, precipitated, and washed as described for chain A. The washed pellets were purified as described by RP-HPLC on C18 column using gradient ranging from 20% to 50% solvent B. The same gradient was used to assess the purity of the linear peptide by analytical RP-HPLC. Peptides were quantified by analytical HPLC using chain B of sCon-Ins G1 as a reference peptide. Out of 108 mg of the resin, 1.31 mg of [B9C(Acm); B21C(Npys)]Con-Ins T1A, 0.82 mg of [B9C(Acm);B21C(Npys)]Con-Ins T1B out of 81 mg, 0.19 mg of [B13C(Acm)]Con-Ins K1 out of 75 mg and 0.26 mg [B12C(Acm);B24C(Npys)]Con-Ins K2 out of 92 mg of the resin. The mass of the peptides was confirmed by electrospray ionization (ESI)-MS.

| Peptide | Purity | Yield | Calculated Monoisotopic [M] | Determined Monoisotopic[M] |
|---|---|---|---|---|
| [B9C(Acm); B21C(Npys)]Con-Ins T1A | 98% | 2.3% | 2,807.14 | 2, 807.13 |
| [B9C(Acm);B21C(Npys)]Con-Ins T1B | 92% | 2.1% | 2,767.09 | 2,766.08 |
| [B13C(Acm)]Con-Ins K1 | 84% | 0.35% | 3127.55 | 3127.54 |
| [B12C(Acm);B24C(Npys)]Con-Ins K2 | 99% | 0.6% | 3140.00 | 3140.00 |

## First intermolecular disulfide bridge formation

### Method 1

Synthesis of [A7B9Cys(Acm)]-Con-Ins T1A and [A7B9Cys(Acm)]-Con-Ins T1B. 100 nmol of chain A was dissolved in 100 µL of 0.1M Tris HCl, pH 8.5 + 6M GuHCl. Such solution was added to either 125 nmol or 150 nmol of chain B dissolved in 100 uL of the same ligation buffer. The reaction was conducted for 2 hr, diluted to 2.5 mL with 0.1% TFA and purified by RP-HPLC using a preparative C18 Vydac column eluted with a linear gradient ranging from 15% to 45% of solvent B in 30 min at a flow rate 4 mL/min. The purity of sCon-Ins T_dimer was assessed by analytical RP-HPLC using the same gradient as for the semi-preparative purification, at a flow rate 1 mL/min. Peptides were quantified at 280 nm using ε value of 2,980 $M^{-1} \cdot cm^{-1}$.

### Method 2

Synthesis of [A7B12Cys(Acm)]-Con-Ins K2. 100 nmol of B-chain was dissolved in 100 µL of 0.01% TFA and transferred to 100 nmol A-chain. Then 200 uL of the ligation buffer: 6M urea, 0.2 M $CH_3CO_2NH_4$, pH = 4.56 was added and the reaction was conducted for 1 hr. Next it was diluted with 0.1% TFA and purified by RP-HPLC using a preparative C18 Vydac column eluted with a linear gradient ranging from 20% to 50% of solvent B in 30 min at a flow rate 4 mL/min. The purity of sCon-Ins T_dimer was assessed by analytical RP-HPLC using the same gradient as for the semi-preparative purification, at a flow rate 1 mL/min. Peptide was quantified at 280 nm using ε value of 2,980 $M^{-1} \cdot cm^{-1}$.

### Method 3

Synthesis of [A7B13Cys(Acm)]-Con-Ins K1 in the presence of 100 nM $CuCl_2$

Combined 100 nmol chain A and chain B of Con-Ins K1 were dissolved in 50% ACN in 0.01% TFA and 100 uL of 1M Tris buffer with 20 nM EDTA pH = 7.5 and 800 uL of $CuCl_2$ solution (125 nM) was added. The reaction was gently stirred and left to proceed for 60 hr. The reaction was diluted with buffer A and purified by RP-HPLC using a preparative C18 Vydac column eluted with a linear gradient ranging from 20% to 50% of solvent B in 30 min at a flow rate 4 mL/min. Peptides were quantified at 280 nm using ε value of 4,470 $M^{-1} \cdot cm^{-1}$. The identity of the peptide was confirmed by ESI-MS.

The identities of the peptides were confirmed by ESI-MS. Yields provided for all methods were calculated based on the starting amount of A-chain.

| Peptide | Purity | Yield | Calculated Monoisotopic[M] | DeterminedMonoisotopic[M] |
|---|---|---|---|---|
| [$^{A7B9}$Cys(Acm)]-Con-Ins T1A | 93% | 36.6% | 5151.05 | 5151.07 |
| [$^{A7B9}$Cys(Acm)]-Con-Ins T1B | 96% | 29% | 5111.00 | 5111.03 |
| [$^{A7B13}$Cys(Acm)]-Con-Ins K1 | 97% | 23% | 6155.63 | 6155.59 |
| [$^{A7B12}$Cys(Acm)]-Con-Ins K2 | 90% | 56% | 5970.53 | 5970.51 |

## Second intermolecular disulfide bridge formation: Iodine ($I_2$) assisted formation of fully folded Con-Ins T1 and Con-Ins K

A solution of $I_2$ (Acros Organics, Geel, Belgium) was prepared as follows: 10 mg of $I_2$ was added to 5 mL of ACN. After 20 min of stirring, the $I_2$ was completely dissolved, and 15 mL of water and 600 µL of TFA were added. 250 µL of the iodine mixture was added to 100 nmol insulin-dimer dissolved in 250 µL of 0.1% TFA (final peptide concentration: 200 µM for Con-Ins T1A, Con-Ins T1B and Con-Ins K1; and 500 µM for Con-Ins K2). The reaction was mixed for 5 min or 10 min, quenched with 10 µL of 1 M L-ascorbic acid (Sigma, St. Louis, MO) in water, diluted with 0.1% TFA in water to a total volume of 4.5 mL and purified by RP-HPLC as described for partially folded Con-Ins dimers. The purity of the final products was assessed by analytical C18 RP-HPLC using the same gradient as for the semi-preparative purification, at a flow rate 1 mL/min. Peptides were quantified as described for the partially folded product. The identity of the peptide was confirmed by ESI-MS. Yield of the reaction was calculated based on the staring amount of the partially folded material with one inter-chain disulfide bridge.

| Peptide | Purity | Yield | Calculated Monoisotopic[M] | DeterminedMonoisotopic[M] |
|---|---|---|---|---|
| Con-Ins T1A | 96% | 41% | 5006.95 | 5006.96 |
| Con-Ins T1B | 87% | 11% | 4966.91 | 4966.91 |
| Con-Ins K1 | 93% | 24% | 6013.55 | 6013.49 |
| Con-Ins K2 | 99% | 29% | 5828.45 | 5829.00 |

## Synthesis of Con-Ins T2 and Con-Ins G3

Con-Ins T2 and Con-Ins G3 were was chemically synthesized, purified and oxidized as previously described before (*Menting JG, et al. How insulin engages its primary binding site on the insulin receptor. Nature 493, 241–245 (2013)*). Here we present a general description of the methods.

## A chain synthesis of [Cys(Acm)$^7$]-Con-Ins T2, [Cys(Acm)$^7$]-Con-Ins G3 and

A-chains of the insulins were synthesized on 0.25 mmol scale using Fmoc strategy on a CEM Liberty one microwave assisted peptide synthesizer (**CEM Corporation,** Matthews, NC), using Fmoc-Cys(Trt) Rink Amide MBHA (0.49 mmol/g load) for [Cys(Acm)$^7$]-Con-Ins T2, and Fmoc-His(Trt) Wang resin (0.32 mmol/g load) (Peptides International, Louisville, KY) for the Con-Ins G3 analogues. The intramolecular disulfide bridge between $^{A6}$Cys and $^{A11}$Cys was formed on the resin following two differnet methods:

For [$^{A7}$Cys(Acm)]-Con-Ins G3 A-chain: deprotection S-*t*-Bu of $^{A6}$Cys with 20% mercaptoethanol (ME) (Fluka,) and 1% *N*-Methylmorpholine (NMM) in dimethylformamide (DMF), preactivation of the thiol with 10-fold excess of 2,2'-dithiobis(5-nitropyridine) (DTNB) ~2.5 mmol (Sigma-Aldrich, St Louis, MO) in dichloromethane (DCM) 30 mL for 1 1/2 hr, followed by the 4-methoxytrityl (Mmt) protecting group removal with 1% trifluoroacetic acid (TFA) in dichloromethane (DCM) 20 mL in the presence of 4 µL triisopropylsilane (TIS) as a scavenger for 25 min to deprotect $^{A11}$Cys(Mmt) and form the disulfide bond between $^{A6}$Cys and $^{A11}$Cys. Subsequently the resin was washed with DMF, DCM, MeOH and dried.

For [$^{A7}$Cys(Acm)]-Con-Ins T2 A-chain: the resin was treated with 1% TFA in DCM (20 mL) in the presence of 2.5% triisopropylsilane (TIS) as a scavenger for 2 min x five times to remove the 4-methoxytrityl (Mmt) protecting groups from $^{A6}$Cys and $^{A11}$Cys. After several washes with DCM and

DMF, 15.6 mg (0.12 mmol) of N-Chloro-succinimide (NCS) reagent in DMF (17 ml) was added to the resin to complete the disulfide bridge formation, after 20 min the resin was washed with DMF, DCM, MeOH and dried.

Cleavage of the peptides from the resin and simultaneous deprotection of the side chains except that of $^{A7}$Cys(Acm) were accomplished by treatment of the resin with TFA/water/TIS: 95/2.5/2.5 or 87.5/5/5/2.5 = TFA/thioanisole/H$_2$O/DOT cleavage cocktail for 2 hr. Cold anhydrous diethyl ether was used to precipitate the crude peptides. (The crude products were purified on Waters PrepPak cartridge (2.5 × 10 cm) packed with Bondapak C$_{18}$(15–20 μm particle size, 300 Å) in solvent system A: 0.1% TFA/water, B: 0.1% TFA/40% water/60% acetonitrile (ACN) with a linear gradient ranging from 25% to 55% solvent B in 60 min at a flow rate 20 mL/min. Fractions containing the right product were pooled and lyophilized. The masses of the peptides were confirmed by ESI-MS.

| Peptide | Purity | Yield | Calculated Monoisotopic[M+H]+ | DeterminedMonoisotopic [M+H]+ |
|---|---|---|---|---|
| [$^{A7}$Cys(Acm)]-con-Ins T2 | 74% | 0.9% | 2417.97 | 2418.24 |
| [$^{A7}$Cys(Acm)]-con-Ins G3 | 99% | 0.9% | 2394.04 | 2394.51 |

## B-chain synthesis of [ $^{B9}$Cys(Acm)]-con-Ins T2 and [$^{B9}$Cys(Acm)]-con-Ins G3

B-chain of Con-Ins T2 and Con-Ins G3 were synthesized in similar manner using Fmoc strategy on a CEM Liberty one microwave assisted peptide synthesizer (**CEM Corporation,** Matthews, NC) on 0.25 mmol scale, using Fmoc-His(Trt) Wang resin (0.32 mmol/g load), and on 0.1 mmol scale, using ChemMatrix Rinkamide resin (0.47 mmol/g load) (PCAS BioMatrix IncSaint-Jean-sur-Richelieu, Quebec Canada J3B 8J8) respectively.

Cleavage of the peptide from the resin and simultaneous deprotection of the side chains except that of $^{B9}$Cys(Acm) were accomplished by treatment of the resin with 87.5/5/5/2.5 = TFA/thioanisole/H$_2$O/DODT cleavage cocktail for 2 hr. Cold anhydrous diethyl ether was used to precipitate the crude peptide. The crude peptide was purified by preparative Waters LC 4000 HPLC on Waters PrepPak cartridge (4.7 × 30 cm) packed with Bondapak C$_{18}$(15–20 μm particle size, 300 Å)(Waters Corporation, Milford, MA) in solvent system A: 0.1% TFA/water, B: 0.1% TFA/40% water/60% acetonitrile (ACN) with a linear gradient ranging from 30% to 50% solvent B in 60 min at a flow rate 100 mL/min. Analytical HPLC screening of the fractions and the determination of the purity of the product were performed on a Phenomenex Kinetex XB-C$_{18}$ column (0.46 × 15 cm, 5 μm particle size, 100 Å pore size)(Phenomenex, Torrance, CA) with a GE AKTApurifier10 analytical HPLC system (GE Healthcare Bio-Sciences, Pittsburgh, PA). The mass of the peptides was confirmed by electrospray ionization (ESI)-MS measured on a ThermoScientific LTQ Orbitrap XL instrument. (Thermofisher Scientific, Waltham, MA)

| Peptide | Purity | Yield | Calculated Monoisotopic[M+H]+ | DeterminedMonoisotopic [M+H]+ |
|---|---|---|---|---|
| [$^{B9}$Cys(Acm)]-Con-Ins T2 | 99% | 18% | 2794.19 | 2794.50 |
| [$^{B9}$Cys(Acm)]-Con-Ins G3 | 96% | 15% | 2571.19 | 2571.77 |

## Con-Ins T2 and Con-Ins G3 inter-molecular disulfide bridge formation between $^{A20}$Cys and $^{B21}$Cys

Peptides chain A and chain B (2.1 μmol of A-chain and 1.9 μmol of B-chain) were directly added to a stirring degassed oxidation buffer of 5 mL DMSO, 5 mL water, 15 mL 0.2 M Tris containing 2 mM EDTA, pH 7.5. The oxidation was monitored by analytical HPLC. After 48 hr at room temperature, the reaction was quenched with 8% formic acid (0.6 mL), diluted with 0.1%TFA to a total volume of 50 mL and purified by preparative HPLC performed on GE AKTApurifier10 HPLC system in solvent system A: 0.1% TFA/water, B: 0.1% TFA/40% water/60% acetonitrile (ACN) with a gradient ranged from 30% to 50%B in 60 min on a Phenomenex Kinetex XB-C$_{18}$ column (2.12 cm x 10 cm, 5 μm particle size, 100 Å pore size) at a flow rate 10 mL/min. Similar method was used to produce Con-Ins G3. Provided yield was calculated based on the starting amount of B-chain.

| Peptide | Purity | Yield | Calculated Monoisotopic[M+H]+ | DeterminedMonoisotopic [M+H]+ |
|---|---|---|---|---|
| [$^{A7B9}$Cys(Acm)]-Con-Ins T2 | 96% | 12% | 5209.13 | 5210.02 |
| [$^{A7B9}$Cys(Acm)]-Con-Ins G3 | 95% | 13% | 4963.21 | 4963.17 |

## Second intermolecular disulfide bridge formation: I$_2$ assisted oxidation to form to fully oxidized Con-Ins T2 and Con-Ins G3

1.1 mg (0.21 µmol) of the partially folded Con-Ins T2 was dissolved in 2.2 mL of 2.5% TFA/water solution and 300 µL of iodine solution (10 mg I$_2$ in 20.6 ml (ACN/H$_2$O/TFA: 5/15/0.6) was slowly added to it, the mixture was stirred for 15 min. The reaction was quenched by adding 1M ascorbic acid solution until the yellow color of the solution became clear. After diluting with 4 mL water, it was purified under the same conditions as the partially folded peptide except with a gradient ranged from 30% to 60%B in 60 min. Purity of the peptide was assessed by analytical HPLC and capillary electrophoresis (CE), which was performed using a Groton Biosystems GPA 100 instrument. The electrophoresis buffer was 0.1 M sodium phosphate (15% ACN), pH 2.5. Separation was accomplished by application of 20 kV to the capillary (0.75 µm x 100 cm). Yield of the reaction was calculated based on the staring amount of the partially folded material with one inter-chain disulfide bridge. Similar method was used to produce Con-Ins G3. For the purposes of the assays peptides were quantified using UV/Vis spectrophotometer at 280 nm using ε value of 1,490 M$^{-1}$·cm$^{-1}$ for Con-Ins G3 and 6,990 M$^{-1}$·cm$^{-1}$ for Con-Ins T2.

| Peptide | Purity | Yield | Calculated Monoisotopic[M+H]+ | DeterminedMonoisotopic [M+H]+ |
|---|---|---|---|---|
| Con-Ins T2 | >98% | 34% | 5065.04 | 5066.07 |
| Con-Ins G3 | 97% | 38.5% | 4918.21 | 4919.07 |

