## [Decision Letter]

Thank you for submitting your article "Fish-hunting cone snail venoms are a rich source of minimized ligands of the vertebrate insulin receptor" for consideration by *eLife*. Your article has been reviewed by three peer reviewers, including Pierre De Meyts as the Reviewing Editor and Reviewer #1, and the evaluation has been overseen by a Reviewing Editor and Mark McCarthy as the Senior Editor. The following individuals involved in review of your submission have agreed to reveal their identity: Ken Siddle (Reviewer #2); Jiri Jiracek (Reviewer #3).

The reviewers have discussed the reviews with one another and the Reviewing Editor has drafted this decision to help you prepare a revised submission.

This is a very interesting study building on the expertise of this group on the biology of venoms from fish hunting cone snails, including the previous demonstration of a shortened insulin-like peptide with unexpectedly high affinity for the fish and human insulin receptors. It has been a tenet in insulin's structural biology (including from this reviewer) that the C-terminal octapeptide of insulin's B-chain is essential for insulin receptor binding and activation. Yet an insulin-like peptide from *Conus Geographus* entirely lacking the end of the B-chain was able to bind to and activate the human insulin receptor. Crystal structure revealed a structural rearrangement where other side chains filled up for the missing key residues. In this paper, the authors report on finding more shortened insulin-like peptides in three *Conus* species with shortened B-chains and great sequence variability, still able to activate the fish and human insulin receptors and cause hypoglycaemia in fish and mice models.

New snail insulins lower blood glucose in zebrafish with potencies comparable to human insulin. Cone insulins bind to human IR-B with potencies ranging between 15-35% of binding potency of human insulin and they also activate Akt phosphorylation by this receptor and reduce blood glucose in mice in vivo. MD simulations revealed that the mode of binding to human insulin mini-receptor construct is similar to previously published complex of the same receptor with Con-Ins G1. Briefly, the interactions of crucial B-chain C-terminus of insulin, which is lacking in snail insulin, are mimicked by aromatic or aliphatic residues from the central B-chain helix of snail insulin. These residues point to the same binding pocket as PheB24 and help to rescue binding affinity. The authors propose that cone snail insulins with their alternative modes of binding to human receptor could represent a new platform for design of new fast acting insulin for treatment of diabetes.

The paper is solid, well conceived and well written, we have no major issues with it and think it will be of great interest to scientists involved in the structural biology and evolution of insulin-like peptides. The experimental work has been very well done and the results are presented very clearly and provide a coherent story of considerable interest on several levels, not least the therapeutic implications.

We have only comments requiring minor modifications.

1) It would be of interest to include in Figure 1 the sequences of endogenous Cone snail insulins, where available, in addition to human and Zebrafish insulins.

2) It would be helpful in Figure 1 to annotate in some way key residues in human insulin that have been implicated in receptor binding (including the B-chain FFY motif). The Discussion section, which focusses very much on the role of the FFY motif, might usefully comment on the conservation or otherwise in cone snail venom insulins of other residues implicated in binding of vertebrate insulins to their receptors.

3) In Figures 3 and 4, the authors should plot on the horizontal axis, and discuss in the text, the actual nM values for ligand concentrations, and nM values for IC50 rather than Log IC50 (they use nM for EC50), to facilitate comparisons.

4) It is noted (subsection “Binding of venom insulins to the hIR induces downstream insulin signalling”) that there is a broad correlation between potencies for receptor binding and activation 'with some differences'. Those differences appear to be that T2 and K1 have significantly higher and G3 significantly lower activation potency than would be expected from their receptor binding (when compared to human insulin). These differences should be highlighted and discussed in relation to the respective sequences.

5) It is noted (subsection “Binding of venom insulins to the hIR induces downstream insulin signalling”) that all but K2 and G3 have significantly higher activation potency than hDOI, although all venom insulins lack the FFY motif. Is it possible to say that K2 and G3 lack surrogates for the FFY motif that are used by other venom insulins to enhance receptor binding?

6) The experiment to test glucose lowering in Zebrafish utilises injections of 65ng venom insulin per g body wt. In terms of the potential role of venom insulins in inducing hypoglycaemic shock to facilitate prey capture (Discussion section), it would be interesting to know the total amount of venom insulin present in cone snails, and the maximum local concentration that might be achieved by release of venom insulin under natural conditions.

7) How did the authors create the sequence logo in Figure 1? Just by drawing in Corel or using some specific software taking into account the real distribution of amino acids at specific positions? In fact, I have some doubts if the figure is correct; e.g. A8 His is present four-times but is small etc.

8) Have the authors some data about cone insulin stabilities in human plasma compared with human insulin? It is not excluded that these venom insulins could be substantially more stable.

9) Add the data for human insulin to Figure 2.

10) The comparison of receptor binding data (Figure 3) with abilities to activate Akt phosphorylation (Figure 4) could indicate some discrepancy. Binding affinities are 15-36% of human insulin but Akt phosphorylation abilities are 0.5-12% of human insulin. This could show on a partial antagonism. Do the authors think that this disproportion is relevant? In our experience, measuring of phosphorylated Akt is sometimes tricky and can be affected by other factors than only by binding of the ligand to the receptor. I would recommend measuring direct IR autophosphorylation as well and comparing the data.

11) If possible, I would appreciate if Figure 6 was bigger e.g. could each panel be one column large? I have problems seeing tiny differences between structures in the printed version.

12) Discussion section: "occupied by the aforementioned aromatic triplet" is not correct. TyrB15 mimics position of human insulin PheB24 only but not PheB25-TyrB26.

13) Appendix, subsection “PEPTIDE SYNTHESIS”: provide synthetic yields as well.

14) Appendix, subsection “First intermolecular disulfide bridge formation”. Individual reactions in Methods 1-3 were done at 100 nmolar scale. It means that roughly 0.3 mg of single chain was used. It is very small scale. Why? The reaction does not proceed well in bigger quantities? How many times was it necessary to repeat the reactions to get sufficient amount of material?

15) Appendix, subsection “Second intermolecular disulfide bridge formation: Iodine (I2) assisted formation of fully folded Con-Ins T1 and Con-Ins K”. The information about peptide quantity in the reaction is missing.

16) In further work the authors may want to study in more detail the kinetics of receptor binding of these peptides and for example if they trigger the negative cooperativity upon receptor binding or instead antagonize it, which may have interesting functional implications. Also, to investigate other biological effects than hypoglycaemia, e.g. mitogenic signalling.

---

## [Author Response]

[…] 1) It would be of interest to include in Figure 1 the sequences of endogenous Cone snail insulins, where available, in addition to human and Zebrafish insulins.

We have added the sequence of the endogenous signaling insulin from *C. geographus* to the bottom of the sequence alignment and added a sentence to the manuscript highlighting distinct differences between the signaling and venom insulin (subsection “Identification and analysis of venom insulin sequences”: “Notably, venom insulins are distinct from the endogenous signaling insulin expressed in the circumoesophageal nerve ring of cone snails (Figure 1C) (Safavi-Hemami et al., 2016). This is also evident by the presence of a distinct signal sequence (and distinct 5’ and 3’ untranslated regions) between the venom insulin gene family and their endogenous homologs (data not shown).”.

2) It would be helpful in Figure 1 to annotate in some way key residues in human insulin that have been implicated in receptor binding (including the B-chain FFY motif). The Discussion section, which focusses very much on the role of the FFY motif, might usefully comment on the conservation or otherwise in cone snail venom insulins of other residues implicated in binding of vertebrate insulins to their receptors.

We have highlighted residues of human insulin known to bind to site 1 or 2 of the human insulin receptor in the revised Figure 1 (Figure 1 caption: “Residues of human insulin involved in binding to sites 1 and 2 of the insulin receptor are indicated by a single (site 1) or double (site 2) green rectangle below the hIns sequence (De Meyts 2016).”

Additionally, we have added several sentences to the revised discussion on other residues that may be of importance for vertebrate IR activation in venom insulins; Discussion section: “Comparative sequence alignment of a diverse set of venom insulin sequences allows for the interrogation of other residues that may play a role for vertebrate receptor activation but that may not easily identifiable by homology modeling. […] This could include Lys/Arg9 and γ-carboxylated Glu4 in the A chain and Glu/Asp17 in the B chain.”

3) In Figures 3 and 4, the authors should plot on the horizontal axis, and discuss in the text, the actual nM values for ligand concentrations, and nM values for IC50 rather than Log IC50 (they use nM for EC50), to facilitate comparisons.

We have chosen to retain the x-axis of the graphs shown in Figure 3 and Figure 4 as these allow easy comparison with the curves published previously reporting cone snail insulin IR binding and activation (Con-Ins G1, (Menting et al., 2016)). However, we have now provided nM IC50 and EC50 values in the text and the revised Figure 3 and Figure 4 to allow easy comparison with insulin and insulin analogue affinities reported in the literature.

While revising the figures we noted an error in our previous Akt graphs (Figure 3). The highest concentration used for hIns and all venom insulins was 860 nM and not 3.44 μM as previously plotted. We regret this error and have revised Figure 3 to now show accurately plotted values and EC50s.

4) It is noted (subsection “Binding of venom insulins to the hIR induces downstream insulin signalling”) that there is a broad correlation between potencies for receptor binding and activation 'with some differences'. Those differences appear to be that T2 and K1 have significantly higher and G3 significantly lower activation potency than would be expected from their receptor binding (when compared to human insulin). These differences should be highlighted and discussed in relation to the respective sequences.

We have also noted these differences but do not yet have sufficient data to properly explain these apparent discrepancies. We have added the following sentence to the revised manuscript; subsection “Binding of venom insulins to the hIR induces downstream insulin signaling”: “[…] some differences can be observed. These include that Con-Ins T2 and Con-Ins K1 have higher and Con-Ins G3 lower activation potency than would be expected from their receptor binding potencies. These observations were not further explored in the current study but could indicate biased signaling of some venom insulins following receptor binding and/or partial receptor antagonism.”

5) It is noted (subsection “Binding of venom insulins to the hIR induces downstream insulin signalling”) that all but K2 and G3 have significantly higher activation potency than hDOI, although all venom insulins lack the FFY motif. Is it possible to say that K2 and G3 lack surrogates for the FFY motif that are used by other venom insulins to enhance receptor binding?

This is an interesting question that is very difficult to answer. In principle, it is possible that Con-Ins G3 and K2 lack surrogates for the FFY motif which would explain their low activation potencies at the hIR. However, there is another likely possibility; these venom insulins may potently activate a member of the insulin receptor family in fish or perhaps even in a specific species of fish, although we are yet to test this directly. We believe that this is a likely scenario since *Conus geographus* and *Conus kinoshitai* are both fish-hunting species and venom insulins specifically evolve to facilitate prey capture (see Safavi-Hemami et al., 2016). Also, when tested in the STZ model in zebrafish, both venom insulins were capable of lowering blood glucose in diabetic fish. Thus, we believe that their low potencies at the hIR may be due to a species-effect rather than a general lack of activity.

6) The experiment to test glucose lowering in Zebrafish utilises injections of 65ng venom insulin per g body wt. In terms of the potential role of venom insulins in inducing hypoglycaemic shock to facilitate prey capture (Discussion section), it would be interesting to know the total amount of venom insulin present in cone snails, and the maximum local concentration that might be achieved by release of venom insulin under natural conditions.

We would like to point out that there is currently no reliable method to accurately measure how much insulin is released into the water. However, we agree with the reviewer that this would be of interest to the reader and have tried to provide an estimate of how much venom might be released.

We have added the following paragraph to the Discussion section: “*Conus geographus* is one of the largest fish-hunting cone snail species (shell length of ~15 cm) that can produce approximately 50 mg of venom in its long and convoluted venom gland. We have previously determined that venom insulins constitute ~ 1/25 of the total venom of this species, corresponding to ~ 2 mg (Safavi-Hemami, Gajewiak et al. 2015). While it remains to be experimentally determined how much venom is released into the water during each predation event, if all venom were injected, 2 mg of venom insulin would be sufficient to effectively lower blood glucose in ~85,500 zebrafish at the concentration used in this study (65 ng insulin/g body weight; ~23 ng per fish).”

7) How did the authors create the sequence logo in Figure 1? Just by drawing in Corel or using some specific software taking into account the real distribution of amino acids at specific positions? In fact, I have some doubts if the figure is correct; e.g. A8 His is present four-times but is small etc.

The sequence logo was generated using Geneious software. Only venom insulin sequences were used for the logo (the human and zebrafish sequence were excluded, *i.e.*, HisA8 is only present 3 times). We have added the following sentence to the Figure 1 caption: “The sequence logo shows conservation/variability at each position in venom insulins (generated using Geneious vs 11.1.2 software).”

8) Have the authors some data about cone insulin stabilities in human plasma compared with human insulin? It is not excluded that these venom insulins could be substantially more stable.

We have not determined serum or plasma stability for these venom insulins yet but would predict that they are prone to proteolytic cleavage due to the presence of several predicted proteolytic cleavage sites in all venom insulins tested here. However, given their streamlined role in envenomation and their minimized structure, venom insulin may indeed be more stable than native human insulin, and it would be interesting to measure their serum stability in the future. We have added the following sentence to the Discussion section: “Given their streamlined role in prey capture venom insulins may exhibit other advantageous properties that, if uncovered, could inform current drug design efforts. For example, venom insulins may have altered off-rates from the receptor, which would affect the ERK signaling properties and resultant mitogenic activities, or may be more stable in extracellular environments, such as blood. Additionally, it would be interesting to determine if venom insulins lack the negative cooperativity observed for human insulin upon receptor binding (De Meyts et al., 1978)”.

9) Add the data for human insulin to Figure 2.

We previously established that injection of 65 ng/g of human insulin lowers blood glucose to 92.0 ± 17.4 mg/dL. We have replotted these values in Figure 2 but emphasize that these were taken from our previous study (subsection “Venom insulins reduce blood glucose in a zebrafish model of diabetes”): “For comparison, 65 ng of human insulin (hIns) /g body weight reduces blood glucose to 92.0 ± 17.4 mg/dL (Safavi-Hemami, Gajewiak et al. 2015).”, and Figure 2 caption: “Data for human Ins (65 ng/g, n=6) was replotted from (Safavi-Hemami, Gajewiak et al. 2015).”

10) The comparison of receptor binding data (Figure 3) with abilities to activate Akt phosphorylation (Figure 4) could indicate some discrepancy. Binding affinities are 15-36% of human insulin but Akt phosphorylation abilities are 0.5-12% of human insulin. This could show on a partial antagonism. Do the authors think that this disproportion is relevant? In our experience, measuring of phosphorylated Akt is sometimes tricky and can be affected by other factors than only by binding of the ligand to the receptor. I would recommend measuring direct IR autophosphorylation as well and comparing the data.

We agree that it would be advantageous to investigate receptor activation in a more holistic manner and, following the reviewer’s comment, we set out to determine direct IR autophosphorylation for the most potent insulins, Con-Ins G1 and Con-Ins T1A (using the HTRF Phospho-IR β (Tyr1150/1151) kit from Cisbio, see revised Materials and methods section). We have included this data in the Supporting information (Figure 4—figure supplement 1). For these two venom insulins IR phosphorylation appears to follow Akt signaling potencies, where venom insulins are ~ 10-15 times less potent than hIns (EC50 hIns: 9.01 nM; Con-Ins G1: 135.5 nM and Con-Ins T1A: 89.4 nM, 95% CI values provided in Figure 4—figure supplement 1).

However, we found this assay difficult to perform as it was very sensitive to ligand treatment times (very low signal after 20 min and very high signal variability when treatment was shorter than 15 min). Even at the optimized ligand treatment time of 15 min there was high signal variability and the signal for hIns plateaued at the 3 highest concentrations used (black arrows in Figure 4—figure supplement 1). We would therefore like to keep this data in the Supporting Information. An explanation of the limitations of the assay (as experienced in our hands) is described in the Figure caption.

We do not think that venom insulins partially antagonize the receptor but do not have data to support this yet. In future studies we would like to more thoroughly investigate the time course of Akt phosphorylation. Preliminary observations suggest that Akt phosphorylation is more similar to that of human insulin at earlier time points (quenching at 10-15 min post insulin exposure versus 30 min as used in the current study and in Menting et al., 2016).

11) If possible, I would appreciate if Figure 6 was bigger e.g. could each panel be one column large? I have problems seeing tiny differences between structures in the printed version.

We have increased the panel size for Figure 6 to allow easier visibility of the structure and labels.

12) Discussion section: "occupied by the aforementioned aromatic triplet" is not correct. TyrB15 mimics position of human insulin PheB24 only but not PheB25-TyrB26.

This has been corrected to read “occupy space which is otherwise occupied by PheB24 in hIns”.

13) Appendix, subsection “PEPTIDE SYNTHESIS”, provide synthetic yields as well.

Synthetic yields are now provided in the revised Appendix (Tables in subsection “PEPTIDE SYNTHESIS”).

14) Appendix, subsection “First intermolecular disulfide bridge formation”. Individual reactions in Methods 1-3 were done at 100 nmolar scale. It means that roughly 0.3 mg of single chain was used. It is very small scale. Why? The reaction does not proceed well in bigger quantities? How many times was it necessary to repeat the reactions to get sufficient amount of material?

Individual A and B chains of each insulin were prepared on different days and aliquoted at 100 nmol. For sheer practical reasons, several parallel reactions each containing 100 nmol of A and B chain were mixed to form heterodimers. These were then pooled and purified together. We anticipate this method to work on larger than 100 nmol scale but have not tested this in detail.

15) Appendix, subsection “Second intermolecular disulfide bridge formation: Iodine (I2) assisted formation of fully folded Con-Ins T1 and Con-Ins K”. The information about peptide quantity in the reaction is missing.

This information is now provided in the revised Appendix (subsection “Second intermolecular disulfide bridge formation: Iodine (I2) assisted formation of fully folded Con-Ins T1 and Con-Ins K”):” 250 μL of the iodine mixture was added to 100 nmol insulin-dimer dissolved in 250 μL of 0.1% TFA (final peptide concentration: 200 μM for Con-Ins T1A, Con-Ins T1B and Con-Ins K1; and 500 μM for Con-Ins K2).

16) In further work the authors may want to study in more detail the kinetics of receptor binding of these peptides and for example if they trigger the negative cooperativity upon receptor binding or instead antagonize it, which may have interesting functional implications. Also, to investigate other biological effects than hypoglycaemia, e.g. mitogenic signalling.

We completely agree with the reviewer that these would be interesting aspects to explore in future studies and have added a sentence to the end of our Discussion section (see response to comment 8 above).